# EFFICIENT MULTI-TURN RL FOR GUI AGENTS VIA DECOUPLED TRAINING AND ADAPTIVE DATA CURATION

## ABSTRACT

Vision-language model (VLM) based GUI agents show promise for automating complex desktop and mobile tasks, but face significant challenges in applying reinforcement learning (RL): (1) slow multi-turn interactions with GUI environments for policy rollout, and (2) insufficient high-quality agent-environment interactions for policy learning. To address these challenges, we propose **DART**, a Decoupled Agentic RL Training framework for GUI agents, which coordinates heterogeneous modules in a highly decoupled manner. DART separates the training system into four asynchronous modules: environment cluster, rollout service, data manager, and trainer. This design enables non-blocking communication, asynchronous training, rollout-wise trajectory sampling, and per-worker model synchronization, significantly improving the system efficiency: *1.6× GPU utilization for rollout*, *1.9× training throughput*, and *5.5× environment utilization*. To facilitate effective learning from abundant samples, we introduce an adaptive data curation scheme: (1) pre-collecting successful trajectories for challenging tasks to supplement sparse success in online sampling; (2) dynamically adjusting rollout numbers and trajectory lengths based on task difficulty; (3) training selectively on high-entropy steps to prioritize critical decisions; (4) stabilizing learning via truncated importance sampling for policy mismatch between policy rollout and updating. On the OSWorld benchmark, *DART-GUI-7B* achieves a 42.13% task success rate, a 14.61% absolute gain over the base model, and 7.34% higher than open-source SOTA. To ensure reproducibility, ***we will fully open-source our training framework, data, and model checkpoints via*** `https://dart-gui-iclr26.github.io` (anonymous), which we believe is a timely contribution to the open-source community of agentic RL training.

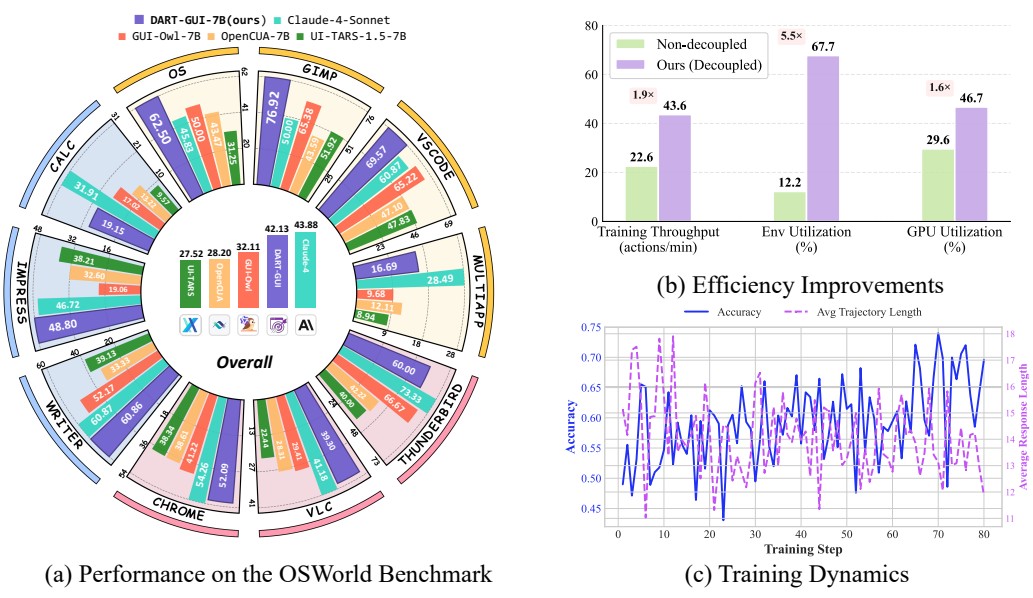

(a) Performance on the OSWorld Benchmark     (b) Efficiency Improvements     (c) Training Dynamics

Figure 1: Overview of the Decoupled Agentic RL Training (**DART**) framework for GUI agents.

# 1 INTRODUCTION

The rapid advancement of large language models (LLMs) (OpenAI, 2025; DeepSeek-AI, 2025) and vision-language models (VLMs) (Bai et al., 2025; Wang et al., 2024; Shen et al., 2025) has accelerated the development of autonomous agents capable of understanding and interacting with graphical user interfaces (GUIs) (Liu et al., 2024; Hong et al., 2024). Such GUI agents (OpenAI, 2025; Bai et al., 2025; Guo et al., 2025b; Fu et al., 2025a) hold significant potential for automating complex desktop and mobile tasks by processing screenshots and natural language instructions. While reinforcement learning (RL) has proven effective for enhancing the reasoning and exploration capabilities of LLMs/VLMs in various domains (Wang et al.; Ye et al., 2025), its application to GUI agents remains particularly challenging. GUI tasks typically involve long-horizon, multi-turn interactions that require maintaining context across dozens of states and actions, making RL training processes prohibitively slow and inefficient.

Recent attempts to apply RL for GUI agents (Lu et al., 2025; Yang et al., 2025; Xi et al., 2025) have yielded only modest improvements (2%–4% on the OSWorld benchmark (Xie et al., 2024)), falling short of closed-source counterparts (Anthropic, 2025b; OpenAI, 2025; Wang et al., 2025a). We identify two primary bottlenecks: First, the tightly-coupled nature of current RL pipelines, where action prediction, environment interaction, data management, and model updates occur sequentially, creates significant idle time, especially given the long episode lengths typical of GUI tasks. Second, the inherent diversity in task difficulty leads to imbalanced learning: agents may overfit to simpler tasks while struggling to explore successful trajectories for more complex ones. Additionally, the sparse reward signals and the presence of critical decision points within long trajectories can introduce noise and instability during training.

To overcome these limitations, we introduce **DART**, a Decoupled Agentic RL Training framework that decouples the RL process into four specialized, asynchronous modules: environment cluster, rollout service, data manager, and trainer. This design enables non-blocking communication and parallel execution, allowing continuous policy updates alongside ongoing environment interactions. By deploying distributed rollout workers and rollout-level trajectory sampling, DART increases resource utilization, achieving a $1.6\times$ improvement in GPU utilization for rollout, a $1.9\times$ increase in training throughput, and a $5.5\times$ boost in environment utilization compared to coupled baselines.

To further enhance the quality and efficiency of learning, we propose an adaptive data curation strategy that operates at multiple granularities. At the *task* and *trajectory* levels, we pre-collect successful trajectories for challenging tasks and dynamically adjust the number of rollouts and maximum trajectory length based on real-time success rates. At the *step* level, we prioritize training on high-entropy steps, identified as critical decision points, within long trajectories. Finally, at the *token* level, we incorporate a truncated importance sampling term to mitigate distribution shift caused by the inference engine and stabilize policy updates. This curated approach ensures that the agent focuses on the most informative experiences, leading to more robust and efficient learning.

We evaluate our method by training *DART-GUI-7B*, a GUI agent model initialized from UI-TARS-1.5-7B (Qin et al., 2025), on the OSWorld benchmark. DART-GUI-7B achieves a 42.13% task success rate, representing a 14.61% absolute gain over the base model and a 7.34% improvement over the previous open-source state-of-the-art. As illustrated in Figure 1, our framework enables stable performance improvement throughout training, even when exploring with shorter trajectories, and demonstrates superior resource efficiency.

Our main contributions are as follows: (1) We propose DART, a novel decoupled RL framework that significantly enhances training efficiency for GUI agents, achieving $1.6\times$ higher GPU utilization for rollout, $5.5\times$ better environment utilization, and $1.9\times$ higher training throughput. (2) We introduce an adaptive data curation scheme that optimizes learning at the task, trajectory, step, and token levels, leading to more effective policy updates. (3) We develop DART-GUI-7B, a state-of-the-art open-source GUI agentic model that achieves superior performance on OSWorld. To promote reproducibility and advance research in agentic RL, we will **fully open-source our training framework, model checkpoints, and curated datasets**.

# 2 RELATED WORK

**GUI Agents** The architecture of GUI agents can be categorized into three primary streams based on how they perceive and interact with user interfaces. Structured agents (Deng et al., 2023; Gur

et al., 2023; Lai et al., 2024) leverage metadata such as APIs or accessibility trees to provide semantic clarity and resilience to layout changes, though their effectiveness is inherently bounded by metadata quality and availability. Visual agents (Xu et al., 2024a; Lu et al., 2024; Cheng et al., 2024) directly process raw screenshots through multimodal LLMs, enabling broader applicability across diverse interfaces but introducing sensitivity to visual variations. Hybrid approaches (Wu et al., 2024; Gou et al., 2024; He et al., 2024) synthesize both modalities, achieving superior grounding performance through complementary information fusion that mitigates the limitations of each individual approach.

The training paradigms for GUI agents have shifted from supervised fine-tuning (SFT)(Lin et al., 2024; Qin et al., 2025; Wang et al., 2025e), which is often constrained by the fixed patterns of demonstration data, to reinforcement learning approaches that learn from environmental feedback. Early RL methods like GUI-R1(Luo et al., 2025) and InfiGUI-R1 (Liu et al., 2025) adopted offline training without real-time interaction, struggling with distribution shift and multi-turn reasoning. Recent online RL frameworks address these limitations through various strategies: ARPO (Lu et al., 2025) extends GRPO for multi-turn interactions, ZeroGUI (Yang et al., 2025) automates task and reward generation via VLMs, while ComputerRL (Lai et al., 2025) designs asynchronous architectures for training API-equipped GUI Agents. Similarly, DistRL (Wang et al.) also utilizes an asynchronous framework, specifically designed for mobile agents with off-policy learning. In comparison, our work focuses on long-horizon desktop tasks by employing a finer-grained decoupling of system modules. This architecture ensures high data freshness, facilitating stable near on-policy training for complex GUI interactions.Our work presents a fully open-sourced reinforcement learning framework specifically designed for GUI agents, integrating the decoupled asynchronous framework with adaptive data curation to achieve both superior performance and community accessibility.

**RL for LLMs, VLMs and Agents**   Reinforcement learning has emerged as a powerful paradigm for enhancing the reasoning and decision-making capabilities of large language models, evolving from preference-based to outcome-based training approaches. While early RLHF methods (Ouyang et al., 2022; Rafailov et al., 2023) relied on expensive human preference annotations that provided only indirect supervision signals, recent advances have shifted toward reinforcement learning with verifiable rewards (RLVR) like GRPO (Guo et al., 2025a) and DAPO (Yu et al., 2025), where automatic and scalable reward signals are derived from concrete task outcomes such as mathematical correctness or code execution success. GSPO (Xu et al., 2024b) further improves training stability and efficiency through sequence-level policy optimization that better handles the credit assignment problem in long sequences.

The computational demands of RL have driven the development of asynchronous architectures that decouple different training components for improved efficiency. Building on successful applications in game AI (Vinyals et al., 2019; Berner et al., 2019), recent frameworks have adapted asynchronous training for language models: AREAL (Fu et al., 2025b) separates rollout generation from model training to maximize GPU utilization while employing staleness-aware PPO to maintain training stability; ROLL (Wang et al., 2025c) provides a comprehensive RL library supporting multi-model pipelines and flexible resource scheduling for large-scale LLM training; Sky-RL (Cao et al., 2025) optimizes text-based tool-use scenarios by employing an asynchronous pipeline to hide CPU-bound tool execution latency behind GPU inference. These frameworks focus on general text-based or multimodal tasks with relatively dense rewards and shorter interaction sequences. We present a decoupled RL framework specifically designed for GUI agent training, addressing the unique challenges of long-horizon multi-modal interactions.

## 3 DART: Decoupled Agentic RL Training Framework

### 3.1 Formulation

We formulate GUI tasks as a sequential decision-making process. At the time step $t$, given the current visual state $s_t$ (a screenshot of GUI) and the interaction history $h_t = \{(s_{\max(1,t-m)}, t_{\max(1,t-m)}, a_{\max(1,t-m)}), \ldots, (s_{t-1}, r_{t-1}, a_{t-1})\}$ of previous $m$ steps, where $r$ denotes the thought for reasoning and $a$ denotes the action (such as clicking on a specific UI element or entering text), along with the task $\tau$, the agent generates a new thought $r_t$ and an executable action $a_t$. Executing the action $a_t$ leads to a new visual state $s_{t+1}$ (an updated screenshot). This interaction loop continues, with the agent repeatedly observing the environment, producing thought

Figure 2: Overall architecture of our framework. The *Rollout Service* interacts with multiple environments in parallel to generate trajectories, which are managed and delivered to the *Trainer* for policy updates. Updated actors are synchronized back to the *Rollout Service*, enabling scalable and efficient asynchronous learning. Implementation techniques are annotated within the figure.

and actions, and receiving updated observations until either a termination condition is met (*e.g.*, the task is completed or fails) or the maximum number of steps is reached. We parameterize the GUI agent using a policy model $\pi_\theta$ (*i.e.*, an VLM) that generates thoughts and actions based on the current state and historical context: $r_t^*, a_t^* = \arg\max_{r_t, a_t} \pi_\theta(a_t | \tau, h_t, s_t)$, where $r_t^*$ and $a_t^*$ represents the optimal thought and action selected by the policy model.

## 3.2 ARCHITECTURE

Our RL framework contains four decoupled modules: *Trainer*, *Data Manager*, *Env Cluster*, and *Rollout Service*, where none of them will be blocked by other modules, as shown in Figure 2. We set up hundreds of real desktop environments in *Env Cluster*, and design a *Rollout Service* to load multiple policy models. The *Env Cluster* receives a sequence of tasks from the *Data Manager*, and samples $N$ trajectories for each task. The *Rollout Service* dynamically assigns idle workers to produce thoughts and actions for different rollouts in parallel. We store sampled trajectories and corresponding rewards in the *Data Manager*. When $N$ trajectories of one task are finished, the *Data Manager* filters and passes them to the *Trainer* based on predefined rules for policy updates. Finally, the updated policy model is synchronized back to the *Rollout Service*, enabling scalable and highly efficient asynchronous learning. Key interactions among these modules are as follows.

## 3.3 ASYNCHRONOUS TRAINER

To improve GPU and environment utilization, we decouple the *Trainer* from the trajectory rollout process, which avoids the blocking between training and rollout. The *Trainer* operates asynchronously, receiving filtered trajectories from the *Data Manager* and performing step-wise GRPO updates. The updated model weights are synchronized to the *Rollout Service*, enabling continuous training while new trajectories are sampled simultaneously.

**Step-wise GRPO.** We adopt step-wise Group Relative Policy Optimization (GRPO) (Shao et al., 2024) to train our GUI agent. For each task $\tau$, we sample $N$ trajectories $T_1, T_2, \ldots, T_N$ using the current policy $\pi_{\theta_{\text{old}}}^{\text{Rollout}}$ of the *Rollout Service*, where the $i$-th trajectory has length $L_i$ and consists of state-thought-action pairs: $T_i = \{(s_{i,j}, r_{i,j}, a_{i,j})\}_{j=1}^{L_i}$. Each trajectory receives a reward $R_i$, and we decompose each trajectory into individual steps and group all steps from the same task for advantage computation. Specifically, we create a step group $\mathcal{D} = \{(h_{i,j}, s_{i,j}, r_{i,j}, a_{i,j}, R_i) \mid i \in [1, N], j \in [1, L_i]\}$, where each step $(s_{i,j}, r_{i,j}, a_{i,j})$ is combined with its history $h_{i,j}$ and the trajectory-level reward $R_i$. We denote "$h_{i,j}, s_{i,j}, r_{i,j}, a_{i,j}$" as "$h, s, r, a$" for simplicity. The step-wise GRPO objective is formulated as

$$\mathcal{J}(\theta) = \mathbb{E}_{(h,s,a,R)\sim\mathcal{D}} \left[ \nabla_\theta \min \left( \frac{\pi_\theta^{\text{Train}}(a|h,s)}{\pi_{\text{old}}^{\text{Train}}(a|h,s)} A, \text{clip}\left( \frac{\pi_\theta^{\text{Train}}(a|h,s)}{\pi_{\text{old}}^{\text{Train}}(a|h,s)}, 1 - \epsilon_{\text{low}}, 1 + \epsilon_{\text{high}} \right) A \right) - \beta \, D_{\text{KL}}\big(\pi_\theta^{\text{Train}}(a|h,s) \,\|\, \pi_\theta^{\text{Ref}}(a|h,s)\big) \right],$$

(1)

where the advantage is computed as

$$A_{i,j} = \frac{R_i - \bar{R}}{\sigma_R}, \quad \bar{R} = \frac{1}{|\mathcal{D}|} \sum_{(h,s,a,R)\in\mathcal{D}} R, \quad \sigma_R^2 = \frac{1}{|\mathcal{D}|} \sum_{(h,s,a,R)\in\mathcal{D}} (R - \bar{R})^2.$$

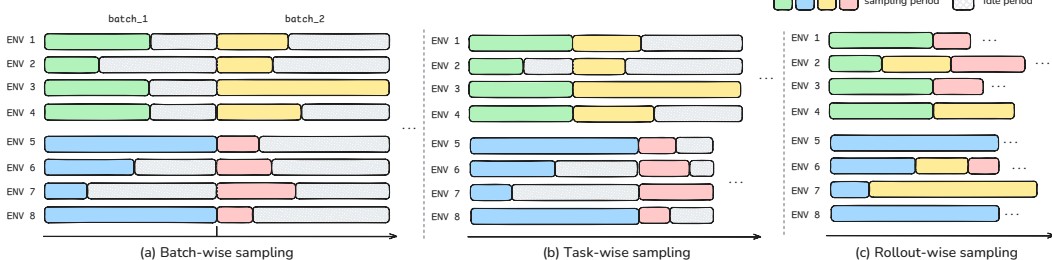

Figure 3: Visualization of sampling timelines for 4 tasks in two batches (denoted by different colors) with rollout number $N = 4$, batch size $= 2$, and a total of 8 environments. Each bar represents the timeline of rollout execution of a task on one environment.

### 3.4 ROLLOUT-WISE SAMPLING

Solving practical GUI tasks (such as OSWorld) typically consists of dozens of steps and lasts for tens of minutes. Thus, sampling efficiency often becomes a major bottleneck. When sampling, tasks within the same batch may vary significantly in difficulty, and even for the same task, minor environmental variations across different executions may produce trajectories of vastly different lengths(Wang et al., 2025d). Under conventional batch-wise sampling (Figure 3 (a)), this heterogeneity causes substantial resource underutilization: environments that complete early remain idle while GPUs stay underused until all trajectories in the batch finish. Task-wise sampling (Figure 3 (b)) partially addresses this issue but still requires waiting for entire tasks to complete before resuming new sampling, limiting overall efficiency. We implement rollout-wise sampling ( Figure 3 (c)), where an individual trajectory serves as the minimal scheduling unit. Once an environment completes a rollout, it immediately launches the next sampling request without waiting for others. This fine-grained scheduling significantly improves environment utilization and maximizes GPU throughput.

Furthermore, rather than statically assigning environments to fixed rollout workers, our decoupled training framework introduces a dynamic model service pool for load balancing. All GPUs are pooled into the shared *Rollout Service*, with incoming requests being distributed to works based on current device utilization. This design provides two engineering advantages: (1) all environments communicate to the *Rollout Rervice* through a unified interface, simplifying system architecture and coordination; and (2) balanced GPU workloads minimize idle time and bottlenecks, resulting in faster inference and higher overall throughput.

### 3.5 PER-WORKER MODEL SYNCHRONIZATION

Model synchronization presents a critical bottleneck in asynchronous GUI agent RL. Traditional approaches rely on global synchronization: when the *Trainer* completes a training iteration, all rollout workers halt operations and wait for every GPU device to receive the updated weights before resuming sampling. As shown in Figure 4 (a), this creates system-wide downtime where environments sit idle and GPUs remain underutilized during each update.

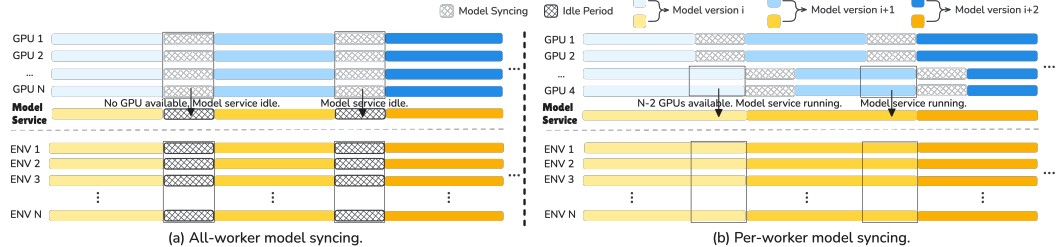

Figure 4: Timeline comparison between all-worker and per-worker model updating with an illustrative example using $N$ GPUs and $M$ environments. The timelines depict idle periods for the GPUs and environments across two model updates, with different model versions represented by varying shades of color. For per-worker model updating, the device number per worker is set to 2.

We introduce per-worker model update to eliminate this bottleneck through staggered parameter distribution. Instead of synchronized model updates across all rollout workers simultaneously, we gradually refresh model weights on rollout workers. It means that when one worker is updating model weights, the others continue serving inference requests with its current model version. Figure 4 (b) illustrates how this maintains continuous service availability: environments never experience complete blocking. This approach delivers two key advantages: dramatically improved sampling throughput through reduced idle time, and seamless model version transitions that preserve ongoing rollout stability.

# 4 MULTI-LEVEL ADAPTIVE DATA CURATION FOR GUI TASKS

## 4.1 PERFORMANCE-AWARE TASK ROLLOUT

Fixed sampling strategies in reinforcement learning waste computational resources by treating all tasks equally. Easy tasks receive excessive sampling while difficult tasks lack sufficient exploration. We propose performance-aware task rollout that dynamically adjusts both sampling frequency and trajectory length based on each task's learning progress.

**Dynamic Rollout Frequency** We continuously monitor each task's success rate and adjust its sampling frequency accordingly. As shown in Figure 5, when a task achieves high success rates (above 0.6), we reduce its rollout frequency from 8 to lower values, preventing overfitting on already-solved tasks. Tasks with low success rates maintain maximum sampling to ensure adequate learning opportunities. This strategy reallocates computational resources from well-learned tasks to challenging ones, improving overall training efficiency.

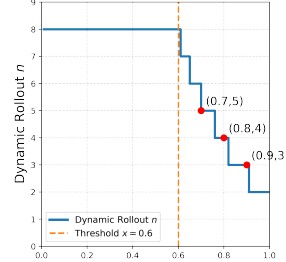

Figure 5: Dynamic rollout $N$ with task success rate.

**Dynamic Trajectory Length** We set task-specific trajectory length limits based on the historical maximum length of successful completions. Instead of using a fixed limit for all tasks, each task receives its own length threshold derived from its successful trajectories. This prevents wasting computation on hopeless long trajectories while allowing sufficient exploration for tasks that genuinely require more steps. For instance, simple clicking tasks might terminate after 10 steps, while complex multi-application tasks can extend to 50 steps. This adaptive approach optimizes the balance between thorough exploration and computational efficiency for individual tasks.

## 4.2 EXPERIENCE POOL OF TRAJECTORIES

Training on challenging tasks poses a significant obstacle in reinforcement learning due to extremely low success rates, which results in insufficient positive trajectories in rollouts. It is common that all trajectories of a task fail, providing no effective learning signal for policy improvement. This severe imbalance between simple tasks and difficult tasks results in training instability, preventing the model from learning correct behavioral patterns. To address this limitation, we introduce an *Experience Pool* that serves as a repository of high-quality successful trajectories for challenging tasks, enabling dynamic supplementation for difficult tasks during training to ensure balanced learning signals.

Specifically, we pre-populate the *Experience Pool* by collecting and storing high-quality successful trajectories through preliminary sampling. During training, when the system detects that all trajectories in the current task fail, it automatically triggers the pool sampling mechanism to randomly retrieve a successful trajectory and incorporate it into the training batch. This design guarantees that every training task contains at least one positive sample while maintaining a reasonable balance between exploration and experience replay, thereby preventing performance degradation on challenging tasks and improving overall training stability.

## 4.3 HIGH-ENTROPY-DRIVEN STEP OPTIMIZATION

Inspired by Wang et al. (2025b), who showed that training exclusively on high-entropy tokens that "act as critical forks that steer the model towards diverse reasoning pathways" can drive effective reinforcement learning, we extend this insight to multi-turn GUI agent training. Low entropy means

non-critical steps for a GUI task, and using such steps may cause instability in training. We design an entropy-based step selection mechanism that identifies and prioritizes the top 80% high-entropy steps for training, thereby encouraging exploration in critical decision-making moments.

For the $t$-th step in a trajectory, we calculate the step-level entropy $H_t$ as the average entropy across all tokens generated in the concatenated thought $r_t$ and action $a_t$ sequence: $H_t = \frac{1}{|r_t|+|a_t|} \sum_{i=1}^{|r_t|+|a_t|} H_{t,i}$, where $H_{t,i} = -\sum_{v=1}^{V} p_{t,i,v} \log p_{t,i,v}$ represents the token-level entropy, with $p_{t,i,v} = \pi_\theta(v|\tau, h_t, s_t, o_{<i})$ being the probability of generating the token $v$. During training, we modify the GRPO objective to include only steps whose entropy is at least larger than 20% steps within the group. This makes reinforcement learning focuses on uncertain steps in GUI navigation.

## 4.4 DISTRIBUTION ALIGNMENT FOR OOD TOKENS

During training, the discrepancy between quantization strategies employed by the *Rollout Service* and the *Trainer* leads to significant differences in their generated policy distributions. In addition, the pre-collected trajectories also has different distributions from the current model. To solve this issue, we follow Yao et al. (2025) and incorporate a truncated importance sampling weight $\min\left(\frac{\pi_{\theta_{\text{old}}}^{\text{Train}}(a|h,s)}{\pi_{\theta_{\text{old}}}^{\text{Rollout}}(a|h,s)}, C\right)$ into the training objective in Eq.(1) to mitigate this gap. By reweighting the gradient contributions based on the probability ratio between the two distributions, we utilize unbiased learning for our decoupled framework, enabling stable training. The final objective is

$$
\begin{aligned}
\mathcal{J}_{\text{HE}}(\theta) = \mathbb{E}_{(h,s,a,R)\sim\mathcal{D}} \Bigg[ \mathbb{I}[H_t \geq \tau_{\mathcal{D}}^{0.2}] \cdot \Bigg( \min\left(\frac{\pi_{\text{old}}^{\text{Train}}(a|h,s)}{\pi_{\text{old}}^{\text{Rollout}}(a|h,s)}, C\right) \cdot \nabla_\theta \min\left(\frac{\pi_\theta^{\text{Train}}(a|h,s)}{\pi_{\text{old}}^{\text{Train}}(a|h,s)} A, \right. \\
\left. \text{clip}\left(\frac{\pi_\theta^{\text{Train}}(a|h,s)}{\pi_{\text{old}}^{\text{Train}}(a|h,s)}, 1-\epsilon_{\text{low}}, 1+\epsilon_{\text{high}}\right) A \right) - \beta\, D_{\text{KL}}\big(\pi_\theta^{\text{Train}}(a|h,s) \,\|\, \pi_\theta^{\text{Ref}}(a|h,s)\big) \Bigg) \Bigg]
\end{aligned}
\tag{2}
$$

where $\mathbb{I}[\cdot]$ is the indicator function, and $\tau_{\mathcal{D}}^{0.2}$ is the threshold larger than 20% entropy in the group.

## 5 EXPERIMENT

### 5.1 SETTINGS

**Experimental Setup**  We evaluate our approach on OSWorld-Verified (Xie et al., 2024), a comprehensive benchmark for assessing multimodal autonomous agents in realistic computer environments. For our training corpus, we adopt the sampling methodology proposed by Lu et al. (2025), selecting a representative subset of 203 tasks from the OSWorld benchmark. We use the results of the evaluation scripts from the OSWorld as rewards for the RL.

**Evaluation Protocol**  We follow the OSWorld evaluation framework (Xie et al., 2024), which uses execution-based validation scripts to assess task completion. Each trajectory receives a reward score in [0, 1] based on programmatic verification of the final system state against predefined success criteria.

**Implementation**  We adopt UI-TARS-1.5-7B (Qin et al., 2025) as the baseline for our policy model. Unlike approaches that rely on multi-agent systems, agent workflows, or agents equipped with additional APIs/tools, we focus on enhancing the capabilities of a single VLM agent through reinforcement learning, aiming to improve the model's inherent decision-making abilities without external scaffolding. Based on decoupled agentic RL training (DART) and the data curation scheme, we obtain the DART-GUI-7B model. More details about training and RL framework can be found in Appendix A.4.

### 5.2 MAIN RESULTS

Table 1 presents results on the OSWorld benchmark across 10 diverse applications. Decoupled Agentic RL Training (DART)-GUI-7B demonstrates well inference efficiency compared to both open-source and closed-source models. DART-GUI-7B achieves 42.13% overall success rate with only 30 maximum steps, establishing a new state-of-the-art among **open-source** models and showing a 12.71% improvement over the baseline UI-TARS-1.5-7B (27.52% with 100 steps). It also achieves comparable performance to Claude-4-Sonnet (41.39% with 100 steps) and outperforming models

Table 1: Results on the OSWorld benchmark. Max Steps indicates the maximum number of agent-environment interactions allowed. **Bold** values denote the best performance among *open-source* models. For brevity, LibreOffice Calc, Impress, and Writer are abbreviated as *calc*, *impress*, and *writer*, respectively. Our results are obtained through evaluation on self-deployed devices using the official codebase and Docker environment. * means results reported in the original papers.

| Model | Max Steps | chrome | gimp | calc | impress | writer | multi_apps | os | thunderbird | vlc | vs_code | Overall |
|---|---|---|---|---|---|---|---|---|---|---|---|---|
| *Closed-source Model* | | | | | | | | | | | | |
| OpenAI CUA o3 (OpenAI, 2025) | 50 | 21.74 | 38.46 | 8.51 | 4.26 | 21.74 | 11.83 | 37.50 | 20.00 | 17.59 | 21.74 | 17.17 |
| OpenAI CUA o3 (OpenAI, 2025) | 100 | 13.04 | 38.46 | 10.64 | 10.64 | 30.43 | 16.53 | 62.50 | 26.67 | 39.18 | 39.13 | 23.00 |
| TianXi-Action-7B (Tang et al., 2025) | 50 | 36.83 | 55.77 | 6.38 | 38.24 | 54.35 | 6.60 | 38.22 | 43.33 | 31.85 | 67.39 | 29.81 |
| OpenAI CUA (OpenAI, 2025) | 100 | 33.61 | 48.08 | 18.09 | 24.47 | 23.91 | 15.61 | 54.17 | 56.67 | 38.21 | 60.87 | 30.50 |
| OpenAI CUA (OpenAI, 2025) | 50 | 36.87 | 34.62 | 14.89 | 29.70 | 26.09 | 15.81 | 70.83 | 66.67 | 11.76 | 69.57 | 31.30 |
| Claude-3-7-Sonnet (Anthropic, 2025a) | 100 | 54.26 | 42.31 | 21.28 | 31.83 | 47.83 | 19.62 | 45.83 | 66.67 | 24.88 | 56.52 | 35.57 |
| Claude-3-7-Sonnet (Anthropic, 2025a) | 50 | 52.09 | 38.46 | 31.91 | 36.09 | 43.48 | 17.66 | 50.00 | 53.33 | 23.53 | 56.52 | 35.83 |
| DeepMiner-Mano-7B (Fu et al., 2025a) | 100 | 39.13 | 69.23 | 27.66 | 42.47 | 56.52 | 17.20 | 50.00 | 73.33 | 35.29 | 78.26 | 40.16 |
| DeepMiner-Mano-72B (Fu et al., 2025a) | 100 | 54.26 | 88.46 | 57.45 | 61.62 | 78.26 | 24.41 | 66.67 | 66.67 | 35.29 | 78.26 | 53.88 |
| Seed1.5-VL-250717 (Guo et al., 2025b) | 100 | 56.52 | 50.00 | 34.78 | 48.91 | 56.52 | 15.35 | 39.13 | 73.33 | 35.29 | 56.52 | 40.18 |
| Claude-4-Sonnet (Anthropic, 2025b) | 100 | 47.74 | 50.00 | 29.79 | 42.47 | 52.17 | 27.86 | 45.83 | 66.67 | 32.82 | 69.57 | 41.39 |
| UI-TARS-250705 (Wang et al., 2025a) | 100 | 56.43 | 50.00 | 40.43 | 55.30 | 60.87 | 14.66 | 41.67 | 66.67 | 44.00 | 52.17 | 41.84 |
| UI-TARS-2-2509 (Wang et al., 2025a) | 100 | 62.96 | 50.00 | 65.96 | 56.38 | 60.87 | 34.13 | 41.67 | 73.33 | 49.94 | 73.91 | 53.11 |
| Claude-4-Sonnet-0514 (Anthropic, 2025b) | 50 | 54.26 | 50.00 | 31.91 | 46.72 | 60.87 | 28.49 | 45.83 | 73.33 | 41.18 | 60.87 | 43.88 |
| Claude-4-Sonnet-0929 (Anthropic, 2025b) | 100 | 62.96 | 53.85 | 72.34 | 68.00 | 82.48 | 49.54 | 70.83 | 60.00 | 58.18 | 73.91 | 62.88 |
| *Open-Source Model* | | | | | | | | | | | | |
| Qwen2.5-VL-32B (Bai et al., 2025) | 100 | 8.70 | 3.85 | 0.00 | 0.00 | 8.70 | 2.15 | 8.33 | 6.67 | 0.00 | 8.70 | 3.88 |
| Qwen2.5-VL-72B (Bai et al., 2025) | 100 | 4.35 | 0.00 | 6.38 | 0.00 | 8.70 | 3.23 | 16.67 | 13.33 | 5.88 | 4.35 | 4.99 |
| ZeroGUI* (Yang et al., 2025) | 15 | - | - | - | - | - | - | - | - | - | - | 20.2 |
| UI-TARS-72B-dpo (Qin et al., 2025) | 50 | 33.24 | 61.54 | 12.77 | 25.45 | 43.48 | 6.71 | 33.33 | 33.33 | 23.53 | 47.83 | 25.88 |
| OpenCUA-7B (Wang et al., 2025e) | 100 | 36.14 | 47.44 | 10.64 | 31.90 | 27.54 | 9.87 | 42.87 | 40.00 | 29.41 | 44.93 | 26.60 |
| UI-TARS-72B-dpo (Qin et al., 2025) | 100 | 32.61 | 73.08 | 6.38 | 23.81 | 34.78 | 8.29 | 37.50 | 60.00 | 17.65 | 52.17 | 26.84 |
| UI-TARS-1.5-7B (Qin et al., 2025) | 50 | 32.79 | 53.85 | 8.51 | 39.31 | 41.30 | 8.60 | 25.54 | 46.67 | 24.41 | 52.17 | 27.25 |
| UI-TARS-1.5-7B (Qin et al., 2025) | 100 | 38.34 | 51.92 | 9.57 | 38.21 | 39.13 | 8.94 | 31.25 | 40.00 | 22.44 | 47.83 | 27.52 |
| OpenCUA-7B (Wang et al., 2025e) | 50 | 38.61 | 43.59 | 13.22 | 32.60 | 33.33 | 12.11 | 43.47 | 42.22 | 28.31 | 47.10 | 28.20 |
| ARPO* (Lu et al., 2025) | 15 | - | - | - | - | - | - | - | - | - | - | 29.9 |
| GUI-Owl-7B (Ye et al., 2025) | 15 | 41.22 | 65.38 | 17.02 | 19.06 | 52.17 | 9.68 | 50.00 | **66.67** | 29.41 | 65.22 | 32.11 |
| GUI-Owl-7B (Ye et al., 2025) | 30 | 41.22 | 73.08 | 12.77 | 25.39 | **60.86** | 10.75 | 50.00 | **66.67** | **47.06** | **69.57** | 34.79 |
| OpenCUA-32B (Wang et al., 2025e) | 50 | 43.39 | 69.23 | 13.48 | 38.28 | 40.58 | 14.93 | 53.62 | 53.33 | 25.49 | 55.07 | 34.14 |
| OpenCUA-32B (Wang et al., 2025e) | 100 | 39.77 | 66.67 | 18.44 | 37.60 | 36.23 | 16.21 | 55.07 | 46.67 | 33.33 | 63.31 | 34.79 |
| *Baseline* | | | | | | | | | | | | |
| UI-TARS-1.5-7B | 30 | 23.91 | 50.00 | 6.38 | 27.45 | 34.77 | 9.54 | 43.48 | 53.33 | 17.65 | 40.91 | 24.17 |
| UI-TARS-1.5-7B | 100 | 38.34 | 51.92 | 9.57 | 38.21 | 39.13 | 8.94 | 31.25 | 40.00 | 22.44 | 47.83 | 27.52 |
| *Ours* | | | | | | | | | | | | |
| DART-GUI-7B | 30 | 52.09 | 76.92 | 19.15 | 48.80 | 60.86 | 16.69 | 62.50 | 60.00 | 39.30 | 69.57 | 42.13 |
| Δ (Ours *vs.* Baseline) | - | ↑13.75 | ↑25.00 | ↑9.58 | ↑10.59 | ↑21.73 | ↑7.75 | ↑31.25 | ↑20.00 | ↑16.86 | ↑21.74 | ↑14.61 |

like OpenAI CUA o3 (23.00% with 100 steps). This significant gain validates the effectiveness of our decoupled asynchronous RL framework and multi-level data curation strategies that focuses on critical decision points. DART-GUI-7B shows consistent improvements across all applications, with particularly strong gains in complex system-level tasks: OS tasks improve by 31.25% (62.50% vs. 31.25%), LibreOffice Writer by 21.73% (60.86% vs. 39.13%), and Thunderbird by 20.00% (60.00% vs. 40.00%). These applications involve longer interaction sequences and diverse action spaces, highlighting our framework's ability to handle long-horizon tasks with sparse rewards effectively.

## 5.3 EFFICIENCY ANALYSIS

We evaluate our decoupled framework's efficiency gains across three key metrics:

1. **Env Util.** ($U_{\text{env}}$): The ratio of time environments spend actively executing actions versus the total available time. $U_{\text{env}} = \frac{\sum_{i=1}^{N_{\text{env}}} t_{\text{active}}^{(i)}}{T_{\text{total}} \times N_{\text{env}}} \times 100\%$, where $t_{\text{active}}^{(i)}$ denotes the active execution time of the $i$-th environment, $N_{\text{env}}$ is the total number of environment instances (80), and $T_{\text{total}}$ is the total wall-clock time of the experiment.

2. **Training Throughput** ($T_{\text{throughput}}$): The average number of action steps collected and used in training per minute. $T_{\text{throughput}} = \frac{N_{\text{actions}}}{T_{\text{total}} \text{ (min)}}$, where $N_{\text{actions}}$ represents the total number of valid action steps used in the training period.

3. **GPU Util.** ($U_{\text{gpu}}$): The ratio of time the GPU(s) spend actively executing computation (e.g., inference and gradient updates) versus the total available time. $U_{\text{gpu}} = \frac{\sum_{j=1}^{N_{\text{gpu}}} t_{\text{active}}^{(j)}}{T_{\text{total}} \times N_{\text{gpu}}} \times 100\%$, where $t_{\text{active}}^{(j)}$ denotes the active computing time of the $j$-th GPU, and $N_{\text{gpu}}$ is the total number of GPUs used in the experiment.

As shown in Table 2, our framework achieves substantial improvements: training throughput nearly doubles from 22.6 to 43.6 actions/min (1.9×), environment utilization increases dramatically from 12.2% to 67.7% (5.5×), and GPU utilization improves from 29.6% to 46.7%(1.6×). These gains

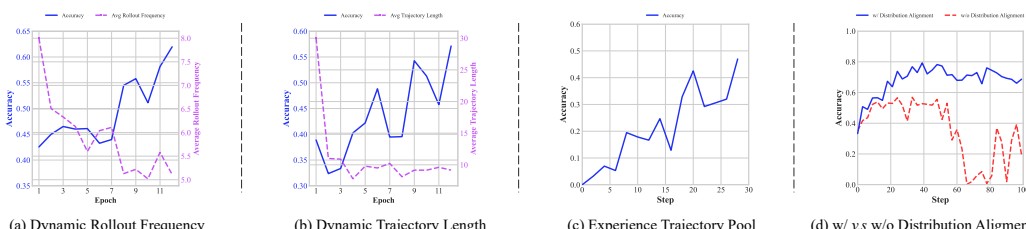

(a) Dynamic Rollout Frequency     (b) Dynamic Trajectory Length     (c) Experience Trajectory Pool     (d) w/ *v.s* w/o Distribution Aligment

Figure 6: (a) Dynamic rollout frequency vs. model accuracy across epochs. (b) Dynamic trajectory length vs. model accuracy over training. (c) Impact of experience trajectory pool on accuracy. (d) Performance comparison with and without distribution alignment.

stem from our decoupled design's elimination of system-wide blocking. Environments continuously generate rollouts without waiting for batch completion, enabling immediate trajectory generation upon task completion. Worker-wise model updates avoid global synchronization, allowing the GPU service to continuously perform inference while some are updating models. This asynchronous operation minimizes idle time across all components, demonstrating that decoupling is essential for efficient GUI agent RL at scale.

Table 2: Efficiency improvements of our decoupled framework compared to non-decoupled baseline.

| System Setup | Training Throughput (actions/min) | Env Util. (%) | GPU Util. (%) |
|---|---|---|---|
| Non-Decoupled | 22.6 | 12.2 | 29.6 |
| Decoupled (Ours) | 43.6 | 67.7 | 46.7 |
| **Improvement** | **1.9×** | **5.5×** | **1.6×** |

Table 3: Ablation study of the data curation scheme. DR stands for dynamic rollout, DTL for dynamic trajectory length, HE for high-entropy-driven step selection, and DA for distribution alignment.

| | Baseline | w/ DR | w/ DTL | w/ HE | w/ DA | Ours |
|---|---|---|---|---|---|---|
| **Pass@1 (%)** | 28.67 | 50.90 | 66.11 | 68.33 | 70.55 | **72.28** |

## 5.4 ABLATION

We conduct ablation studies on a subset of 45 tasks from the training set to evaluate the four levels of our data curation scheme. Baseline means only using the decoupled RL training framework, and Ours means we apply the whole data curation scheme is added.

**Dynamic Rollout and Dynamic Trajectory Length.** We evaluate the impact of our mechanisms on training efficiency. As shown in Table 3, both approaches effectively improves the baseline performance. As shown in Figure 6(a) and (b), both strategies effectively reduce computational overhead as the model improves. Dynamic Rollout demonstrates that as accuracy increases, the average rollout frequency decreases from 8.0 to 5.0 per task. Similarly, Dynamic Trajectory Length shows that as performance improves from, the average trajectory length drops from 30 to less than 10 steps, as the model learns to complete tasks more efficiently. This confirms that our adaptive mechanisms successfully accelerate training by eliminating redundant computation while maintaining exploration on challenging tasks.

**Experience Pool of Trajectories.** We evaluate the impact of the experience pool on training performance for a set of 22 challenging tasks which initially exhibits a success rate of 0%. As illustrated in Figure 6(c), initially the model fails to sample any correct trajectories, resulting in a 0% success rate at the first step. During training, by dynamically incorporating successful trajectories from the pool when all online rollouts fail, the model progressively improves its performance, reaching 46% by the later steps. This demonstrates that the Experience Pool effectively mitigates the sparse positive signal problem and stabilizes learning on tasks with extremely low natural success rates.

**High-entropy-driven Step Selection.** We evaluate the impact of our high-entropy-driven step selection on agent performance. As shown in Table 3, compared to the baseline without high-entropy step prioritization, this approach improves success rates from 28.67% to 68.33% compared to the baseline, demonstrating its effectiveness.

**Distribution Alignment.** We examine the distribution alignment in stabilizing multi-turn VLM RL. As shown in Figure 6(d), incorporating rollout log probabilities as importance weights provides three key benefits: (1) maintains training stability with consistent 70% accuracy throughout training, (2) achieves higher peak performance (78% vs. 55%), and (3) prevents catastrophic collapse that occurs in the baseline, which drops from 55% to near 0% after step 60, a common issue in agent RL. We also observe that our method improves performance over the baseline, reaching 70.55% in Table 3. By reweighting gradients according to the probability ratio between rollout and current distributions, our method effectively mitigates distribution shift, enabling stable learning in long-horizon GUI tasks.

## 5.5 ROBUSTNESS ANALYSIS

We conduct a series of experiments to verify that our gains stem from robust policy learning rather than simple imitation or overfitting.

To distinguish the contribution of our RL framework from simple imitation, we conduct a experiment by training a standard SFT model on 34k successful steps collected from the base model UI-TARS-1.5-7B. As shown in Table 4, while SFT brings a marginal improvement (28.14% vs. 24.17%), it significantly underperforms our RL method (42.13%). This performance gap confirms that our RL gains stem from learning robust interaction policies capable of handling dynamic variations, rather than overfitting to fixed trajectories.

Table 4: Comparison between UI-TARS-1.5-7B (baseline), SFT variant, and DART-GUI-7B on the full OSWorld set (Max steps=30).

| Method | Acc (%) |
|---|---|
| UI-TARS-1.5-7B | 24.17 |
| UI-TARS-1.5-7B-SFT | 28.14 |
| **DART-GUI-7B (Ours)** | **42.13** |

To rule out overfitting to specific training tasks, we evaluate our model on a held-out validation set comprising 171 constructed new tasks (distinct from training) annotated by senior AI researchers. As shown in Table 5, our method achieves 36.06%, significantly outperforming the baseline's 27.19%. This consistent performance on unseen tasks provides strong evidence that our policy has learned robust GUI interaction patterns rather than memorizing specific task solutions.

Furthermore, to rule out overfitting to OSWorld simulator, we extend our evaluation to the UI-Vision benchmark (Nayak et al., 2025), which introduces distinct interface distributions. Evaluated in a strict zero-shot setting without fine-tuning (Table 5), DART-GUI-7B consistently outperforms the baseline. These results confirm that our method captures fundamental GUI interaction capabilities that remain robust across substantial domain shifts, rather than relying on simulator-specific shortcuts.

Table 5: Performance comparison on 171 human-annotated held-out tasks and the UI-Vision benchmark (Zero-shot).

| Method | Held-out Tasks (%) | UI-Vision Benchmark (%) | | | |
|---|---|---|---|---|---|
| | Acc | Basic | Functional | Spatial | Overall |
| UI-TARS-1.5-7B | $27.19 \pm 2.14$ | 20.82 | 21.84 | 7.49 | 16.44 |
| **DART-GUI-7B** | $\mathbf{36.06 \pm 1.38}$ | **21.67** | **22.40** | **8.48** | **17.24** |

## 6 CONCLUSION

In this paper, we have introduced an efficient reinforcement learning (RL) method to address key challenges in training GUI agents powered by vision-language models (VLMs). Our approach overcomes two major limitations in current RL frameworks for GUI tasks. The proposed decoupled RL framework can speed up the training process, and the data curation scheme can improve the quality of training data for multi-turn agent-environment interactions. In this case, we significantly improved both GPU and environment utilization and further ensure better agent performance. Our experimental results on OSWorld demonstrate the efficacy of our approach, achieving a 42.13% task success rate, outperforming existing open-source models and the baseline by substantial margins. The ablation studies highlight the critical role of the data curation scheme in optimizing the RL process. This work presents a promising direction for improving the efficiency and success of RL-based GUI agents and provides valuable insights for future developments in this field.

## ETHICS STATEMENT

We have adhered to the ICLR Code of Ethics throughout this work. Our study does not involve human subjects or sensitive personal data, and all experiments were conducted on simulated desktop environments. We take care to ensure that the methods, datasets, and models are used responsibly, avoiding potential harms related to bias, discrimination, or misuse. Any pretrained models and source code released will follow standard practices for safe distribution. We declare no conflicts of interest or external sponsorships that could have influenced the research outcomes.

## REPRODUCIBILITY STATEMENT

We have made several efforts to ensure that the results reported in this paper are reproducible.

For the proposed decoupled RL framework, we provide detailed descriptions of the overall architecture and algorithmic components in Section 3, and of the adaptive multi-level data curation and exploration strategies in Section 4.

Our experimental setup and evaluation protocols are described in Section 5.1, including dataset configurations, environment settings, and performance metrics. The implementation details of the training framework, such as the Trainer, Rollout Service, Environment Cluster, and Data Manager, are provided in Section 5.1, with additional specifics in Appendices A.1– A.2.

All ablation studies and efficiency analyses are reported in Sections 5.2– 5.4, with implementation details and additional examples provided in Appendices A.3– A.8. These include descriptions of action spaces, system prompts, and extended experimental results, which together provide sufficient information for reproducing the reported performance.

Additionally, source code, configuration files, and pretrained models will be made available upon acceptance to facilitate full reproducibility.

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

# A APPENDIX

## A.1 BROADER IMPACT

Our work advances GUI automation through efficient reinforcement learning, offering benefits in accessibility, productivity, and evaluation. DART-GUI can assist users with disabilities, streamline repetitive tasks, and enable more thorough UI validation. By releasing our framework and models openly, we lower the barrier for researchers and developers while reducing environmental impact through efficient training. At the same time, we recognize risks such as unauthorized access or privacy concerns and emphasize the importance of responsible use with proper safeguards.

## A.2 LLM USAGE STATEMENT

We acknowledge the use of LLMs as a writing assistance tool during the preparation of this manuscript. The LLMs were utilized exclusively for improving language quality, including grammar correction, and enhancing clarity. All scientific contributions, including research conceptualization, methodology, experimental design, data analysis, and interpretation of results were conducted solely by the human authors. The LLMs did not generate any original research ideas, hypotheses, or substantive scientific content. The authors assume full responsibility for the accuracy, integrity, and originality of all content presented in this work, including any portions where language was refined with LLM assistance.

## A.3 SYSTEM PROMPT AND ACTION SPACE

We follow the system prompt of the baseline model UI-TARS-1.5-7B(Qin et al., 2025), as shown in Figure 7.

---

**System Prompt for GUI Agent**

You are a **GUI agent**. You are given a **task** and your **action history**, with **screenshots**. You need to perform the next action to complete the task.

**## Output Format**
```
Thought: ...
Action: ...
```

**## Action Space**
```
click(start_box='<|box_start|>(x1,y1)<|box_end|>')
left_double(start_box='<|box_start|>(x1,y1)<|box_end|>')
right_single(start_box='<|box_start|>(x1,y1)<|box_end|>')
drag(start_box='<|box_start|>(x1,y1)<|box_end|>',
     end_box='<|box_start|>(x3,y3)<|box_end|>')
hotkey(key='')
type(content='') # If you want to submit, use "\n" at the end
scroll(start_box='<|box_start|>(x1,y1)<|box_end|>',
       direction='down or up or right or left')
wait() # Sleep for 5s and take a screenshot
finished(content='xxx') # Use escape characters \', \", \n
```

**## Note**
- Use {language} in Thought part.
- Write a **small plan** and finally **summarize** your next action (with its target element) in one sentence in Thought part.
- My computer's password is 'password', feel free to use it when you need sudo rights.

**## User Instruction**
{instruction}

---

Figure 7: System prompt template for DART-GUI agent with action space definition and output format specifications.

### A.4 MORE IMPLEMENTATION DETAILS

**Trainer.** The training pipeline utilizes Fully Sharded Data Parallel (FSDP) (Zhao et al., 2023) via verl (Sheng et al., 2024) for distributed training across 8 NVIDIA H100 GPUs. The learning rate is set to $1 \times 10^{-6}$ with a KL divergence regularization coefficient of $\beta = 0.1$. Following DAPO (Yu et al., 2025), dynamic clipping boundaries are configured with $\epsilon_{\text{low}} = 0.2$ and $\epsilon_{\text{high}} = 0.28$. The rollout policy scaling parameter is set to $C = 1$.

**Rollout Service.** Model deployment employs vLLM (Kwon et al., 2023) as the rollout service, incorporating load balancing mechanisms to distribute workload across devices. *Worker-wise Model Syncing* is implemented to enable non-blocking operation. Each worker is allocated 2 NVIDIA H100 GPUs. The sampling temperature is set to $1.0$, with a maximum of 30 interaction steps per episode. The initial rollout num is configured as $n_{\text{rollout}} = 8$.

**Env Cluster.** We use Kubernetes(K8s) Burns et al. (2016) to orchestrate 180 parallel Ubuntu Docker containers serving as environment instances. Each environment operates independently, receiving actions from agents and returning screenshots as observations. Figure 8 shows the dashboard of our cluster.

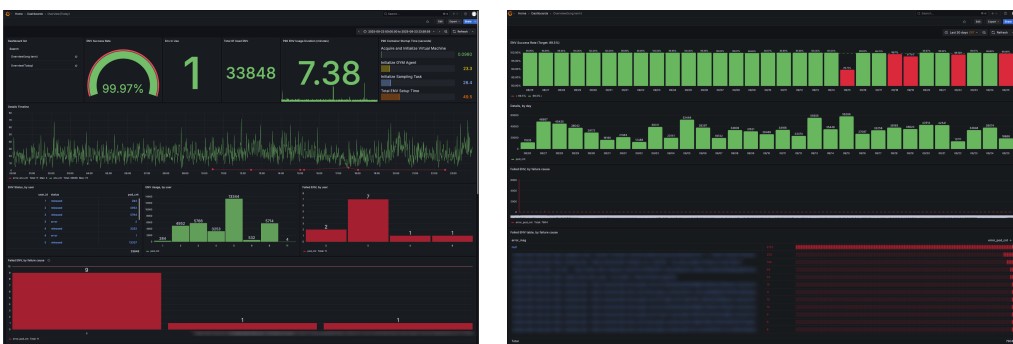

Figure 8: Dashboard of the distributed Ubuntu env cluster.

**Data Manager.** We build a centralized *Data Manager* built on MySQL (Oracle Corporation, 2024) that handles data storage and coordination across the entire training pipeline. The database architecture, illustrated in Figure 9 and summarized in Table 6, comprises 11 interconnected tables organized into four functional categories.

For model management, the *Data Manager* tracks model checkpoints and versions through the `checkpoint`, `current_model`, and `model_registry` tables, enabling seamless model versioning and deployment. The data management subsystem, consisting of `datasets`, `dataset_usage_events`, `rollout_run`, and `rollout_chunk` tables, maintains comprehensive records of trajectories, their usage patterns, and associated rewards. Each trajectory is uniquely identified and linked to its corresponding run, task, and model version.

For the *Trainer*, the *Data Manager* ensures balanced training by monitoring trajectory outcomes through the `reward` field in multiple tables. When sampling training data, the *Data Manager* guarantees each task contains at least one successful trajectory (positive reward) and one failed trajectory (negative or zero reward). If all sampled trajectories for a given task fail, the *Data Manager* queries the `datasets` and `rollout_run` tables to retrieve positive trajectories from previously collected data, using the `trajectory_id` and `task_id` as keys to maintain task consistency. The `dataset_usage_events` table tracks these data access patterns, recording each usage with timestamps and model versions to ensure reproducibility. Additionally, the `trainable_group` table aggregates trajectories ready for training, while the `update_model_task` table manages the model update pipeline, coordinating between checkpoints and deployment stages.

Table 6: Database tables categorized by functionality.

| Category | Table Count | Tables |
|---|---|---|
| Model Management | 3 | checkpoint, current_model, model_registry |
| Data Management | 4 | datasets, dataset_usage_events, rollout_run, rollout_chunk |
| Training | 2 | trainable_group, update_model_task |
| Inference | 2 | inference_node, inference_tasks |
| **Total** | **11** | |

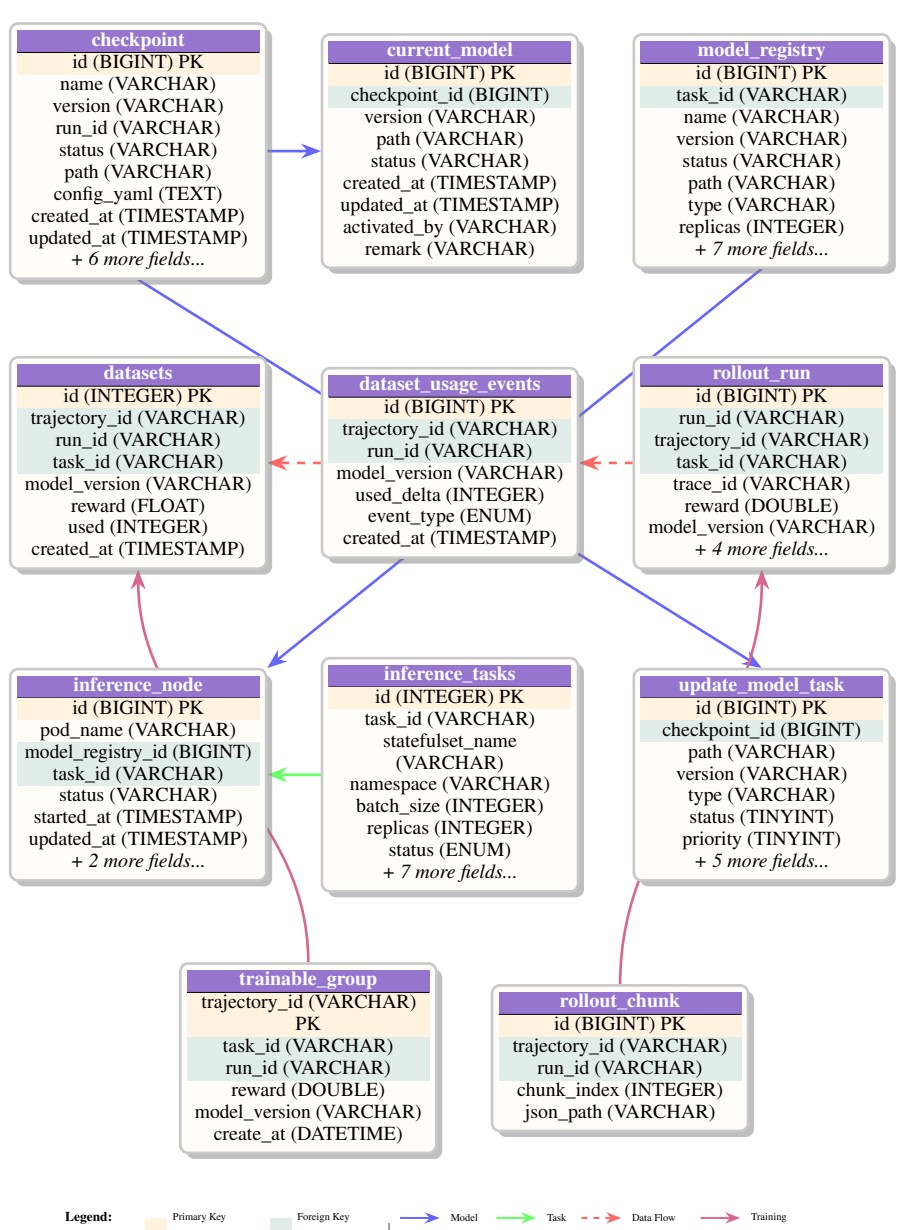

Figure 9: Database Schema Visualization showing the relationships between 11 tables. Golden cells indicate primary keys, green cells indicate foreign keys. Blue arrows represent model management relationships, green arrows show task dependencies, red dashed arrows indicate data flow between rollout and dataset tables, and purple arrows represent training data connections.

## A.5 COMPARE WITH RELATED FRAMEWORKS

In this section, we provide a detailed comparison between DART and other related RL frameworks, highlighting the specific architectural and algorithmic choices driven by the desktop GUI domain.

**Comparison with DistRL.** DistRL (Wang et al.) is a highly relevant baseline that also employs an asynchronous framework with decoupled training. However, the distinction between DART and DistRL is fundamental, driven by the different requirements of desktop GUI versus mobile GUI tasks. As detailed in Table 7, these domain differences lead to key innovations in system architecture, sampling strategy, and training algorithms. DART offers three specific advantages over DistRL beyond implementation details:

1. **Finer-Grained Decoupling for Long-Duration Tasks:** DART introduces a three-way decomposition by separating the `Rollout Service` from the `Environment Cluster`. This design directly addresses the bottleneck of environment availability in desktop GUI tasks, where episodes last 5–20 minutes. In contrast, DistRL's integrated worker model suffices for mobile tasks lasting seconds to minutes. DART's finer decoupling improves the environment utilization by $5.5\times$ and GPU utilization for rollout by $1.6\times$.

2. **High-Freshness Sampling for Near-On-Policy Learning:** DART employs a high-freshness, single-use sampling strategy, where each trajectory is used exactly once and the version gap is constrained to one step. This approximates on-policy learning, avoiding the complexity of off-policy corrections (e.g., Retrace($\lambda$) used in DistRL). This strategy ensures greater training stability and higher throughput ($1.9\times$).

3. **Domain-Optimized Algorithmic Simplicity:** DART uses step-wise GRPO with rule-based rewards, tailored to the long-horizon nature of desktop tasks. This contrasts with DistRL's reliance on LLM-based rewards and learned value networks, which introduce additional complexity.

Table 7: Comparison between DART and DistRL.

| Aspect | DART (Ours) | DistRL |
|---|---|---|
| **Domain** | Desktop GUI | Mobile GUI |
| **Task Time** | 5-20 minutes per episode | Seconds to minutes per episode |
| **Decoupled Modules** | Four Modules: `Trainer`, `Rollout Service`, `Environment Cluster`, `Data Manager` | Two Modules: `Host Learner`, `Worker` (integrated inference + environment) |
| **Samples** | High-freshness, single-use trajectories | Replay-based, reused trajectories |
| **Training Algorithms** | Step-wise GRPO, a nearly **on-policy** RL algorithm | A-RIDE, an **off-policy** RL algorithm with off-policy corrections |
| **Rewards & Advantages** | Rule-based rewards (OSWorld scripts) + Group-normalized advantages | LLM-based rewards (Gemini-1.5-Pro) + Learned value network |

**Comparison with Sky-RL.** We also compare DART with Sky-RL (Cao et al., 2025), a representative framework for text-based tool use. While both systems aim to improve multi-turn RL efficiency, they address fundamentally different bottlenecks. As analyzed in Table 8, Sky-RL focuses on logic-heavy scenarios where CPU-bound tool execution is limiting, employing standard pipelined synchronization to ensure data consistency. In contrast, DART targets the visual-heavy Desktop GUI domain, where the high variance of VLM inference latencies creates synchronization challenges. To address this, DART implements a **per-worker non-blocking synchronization** strategy. This allows workers to stagger updates independently, preventing the system-wide idle time that would otherwise occur during heavy visual processing.

**Empirical Efficiency Comparison.** For empirical performance comparison, we focus on frameworks natively designed for GUI agents: ARPO (Lu et al., 2025) and ZeroGUI (Yang et al., 2025). As shown in Table 9, DART significantly outperforms existing methods:

- **vs. ARPO:** ARPO employs a standard synchronous rollout-training loop, architecturally equivalent to our "Non-Decoupled" baseline. DART achieves a $1.9\times$ speedup and significantly higher accuracy (42.13% vs 29.9%).

Table 8: Comparison of efficiency bottlenecks and architectural optimization strategies between DART and Sky-RL.

| Feature | DART (Ours) | Sky-RL |
|---|---|---|
| Target Domain | Desktop GUI (Visual-heavy) | Code/Research (Text/Logic-heavy) |
| Primary Bottleneck | VLM Inference & Rendering (Encoding high-res screenshots) | Tool Execution Latency (CPU-bound tasks like running tests) |
| Efficiency Strategy | Decoupled & Asynchronous (Separating Trainer/Rollout to maximize VLM throughput) | Async Pipeline (Hiding CPU tool latency behind GPU inference) |
| Model Sync | Per-Worker Non-Blocking Sync (Staggered updates to prevent idle time) | Standard/Pipelined Sync (Optimized for data flow consistency) |

- **vs. ZeroGUI:** ZeroGUI introduces substantial overhead by replacing rule-based verification with an LLM-based reward model, requiring 4x VLM inferences per trajectory. DART achieves superior performance with a lightweight rule-based reward system.

Table 9: Estimated Efficiency and Performance Comparison against ARPO and ZeroGUI.

| Framework | Architecture | Reward Mechanism | Training Throughput | Env Util.(%) | Success Rate(%) |
|---|---|---|---|---|---|
| ARPO | Coupled (Sync) | Rule-based | 22.6 actions/min | 12.2 | 29.9 |
| ZeroGUI | Coupled (Sync) | VLM-based (4x Voting) | 19.4 actions/min | 11.4 | 20.2 |
| DART (Ours) | Decoupled (Async) | Rule-based | 43.6 actions/min | 67.7 | 42.1 |

## A.6 MORE ABLATION

We conduct additional ablation studies on the full OSWorld benchmark to further isolate the contributions of our framework components and verify the robustness of our data strategy.

**Full-Set Framework Ablation.** To verify the impact of the Decoupled Framework and the Distribution Alignment (DA) module beyond the subset used in the main text, we conducted experiments on the full OSWorld benchmark. As shown in Table 10, the results isolate the gains from specific components:

- **Effect of Decoupling:** Even without Distribution Alignment, our decoupled architecture ('Ours w/o DA') achieves 34.77%, significantly outperforming the coupled baseline (24.17%). This confirms the intrinsic benefit of the asynchronous design in improving data collection and throughput.
- **Effect of Distribution Alignment:** Adding the DA module ('Ours w/ DA') boosts performance to 38.81%, validating that correcting distributional shifts is critical for stability at scale.

Table 10: Ablation study on the full OSWorld benchmark (Max Steps=30). "Ours (Full Method)" includes DA + Dynamic Rollout (DR) + Dynamic Trajectory Length (DTL) + High Entropy (HE).

| Setting | Task Success Rate(%) | Improvement(%) |
|---|---|---|
| UI-TARS-1.5-7B (Baseline) | 24.17 | - |
| Ours (Decoupled w/o DA) | 34.77 | +10.60 |
| Ours (Decoupled w/ DA) | 38.81 | +4.04 |
| **Ours (Full Method)** | **42.13** | **+3.32** |

**Impact of Experience Pool.** To address concerns that pre-collecting successful trajectories for hard tasks might lead to memorization, we conducted an ablation removing the Experience Pool entirely. As shown in Table 11, the performance drop is marginal (1.11%), demonstrating that the model can successfully learn robust policies through online sampling alone, albeit with higher computational cost.

Table 11: Ablation study removing pre-collected trajectories (Experience Pool).

| Configuration | Success Rate (%) |
|---|---|
| DART w/o pre-collected trajectories | 41.02 |
| **DART (Full Method)** | **42.13** |

## A.7 VISUALIZATION

**Key Steps of Tasks.** We visualize the comparison between the baseline and our model on several tasks, demonstrating that through our RL training, our model can make correct actions at critical steps, thereby achieving successful trajectories. The comparative analysis clearly shows the improvement in decision-making at pivotal moments, as illustrated in Figure 10 and Figure 11.

**Visualization of Extremely Difficult Tasks.** By leveraging pre-collected successful trajectories from the trajectory pool for extremely difficult tasks (where pass@32 failed), our trained model demonstrates the ability to generate correct trajectories on these challenging tasks. We visualize the critical steps that truly determine the success or failure of trajectories in these tasks and provide detailed analysis of the underlying reasons, as shown in Figure 12 and Figure 13.

## A.8 FAILURE CASES

**Visualization of Failure Cases.** We also visualize representative failure cases to highlight the limitations of our model. These examples demonstrate situations where DART-GUI-7B makes mistakes at key steps, preventing successful task completion, as illustrated in Figure 14.

**Statistics of Failure Cases.** To provide a mechanistic understanding of the model's limitations, we conduct a comprehensive human study on 65 failed tasks from DART-GUI-7B and 55 corrected tasks (where DART succeeded but the baseline failed). We categorize the trajectories into five distinct failure modes:

- **Reasoning Failures**: The agent correctly grounds UI elements but selects an incorrect action or logical sequence (e.g., clicking "Done" before changing a setting).
- **Grounding Errors**: The agent fails to locate the target element or generates hallucinated coordinates.
- **Looping/Stuck**: The agent enters a repetitive action sequence, or the environment remains trapped in a stagnant state (e.g., unclosable pop-ups or unresponsive pages) preventing progress.
- **Knowledge Limitations**: The agent lacks the specific domain knowledge or application usage experience required to solve the task.
- **Action Space Limitations**: The agent generates a valid intent, but the action fails due to API constraints or complex UI interactions (e.g., specific drag-and-drop mechanics).

The detailed statistical breakdown of these error modes is presented in Table 12.

Table 12: Error Distribution Analysis. Comparison of failure modes between tasks failed by the baseline (but corrected by DART) and the remaining failures of DART-GUI-7B.

| Error Category | Baseline Failures (Corrected by DART) | DART-GUI-7B Failures (Remaining Errors) |
|---|---|---|
| Reasoning Failures | 61.82% | 40.00% |
| Grounding Errors | 16.36% | 7.69% |
| Action Space Limitations | 1.82% | 6.15% |
| Looping/Stuck | 1.82% | 7.69% |
| Knowledge Limitations | 18.18% | 38.46% |

We further visualize this distribution shift in Figure 15. The comparison highlights that DART significantly reduces Reasoning Failures (dropping from 61.8% in corrected tasks to 40.0% in

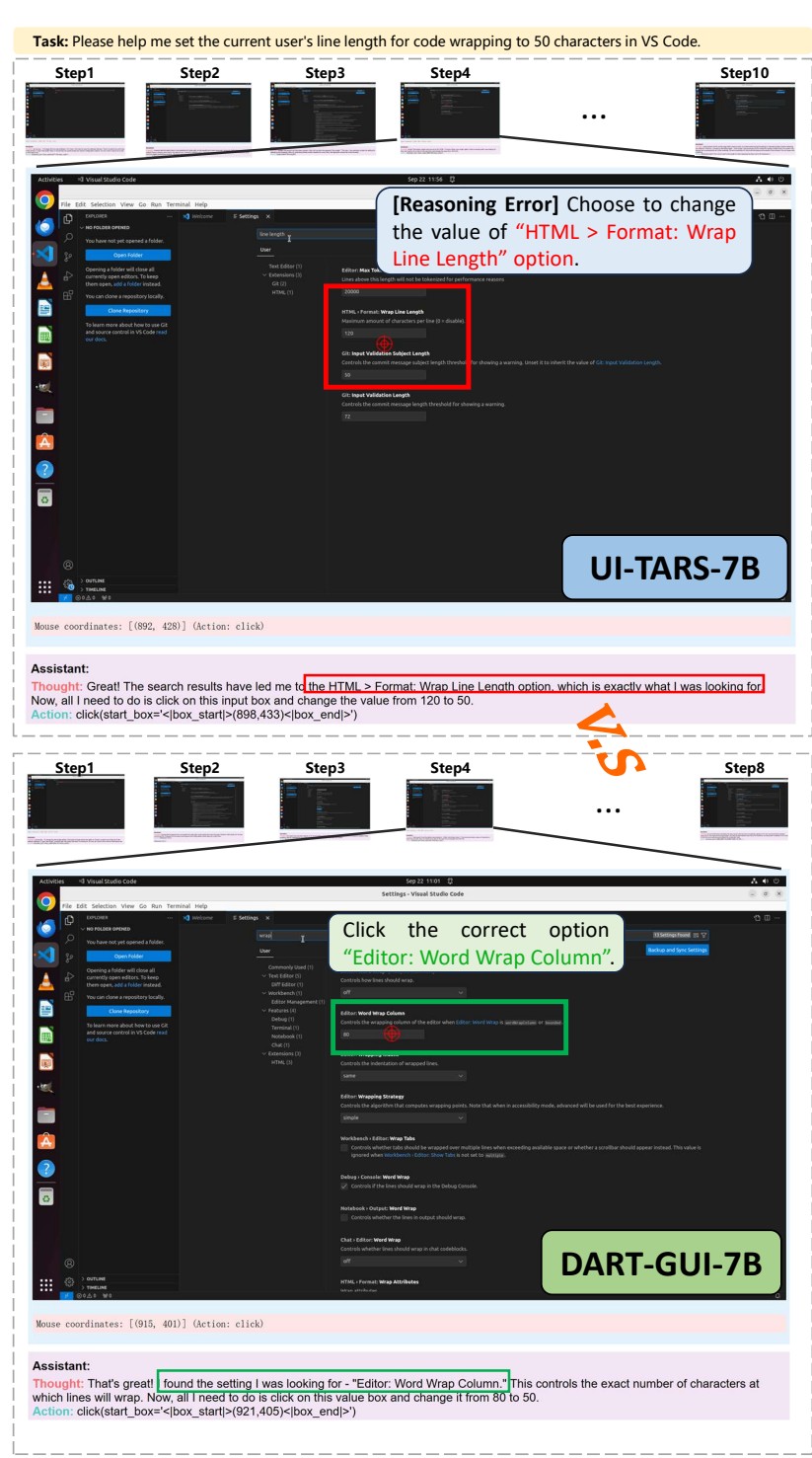

Figure 10: Case study comparing UI-TARS-7B and DART-GUI-7B on configuring line wrapping in VS Code. UI-TARS-7B exhibits a reasoning error by modifying the unrelated HTML > Format: Wrap Line Length option, whereas DART-GUI-7B correctly locates and sets the Editor: Word Wrap Column parameter to the desired value.2

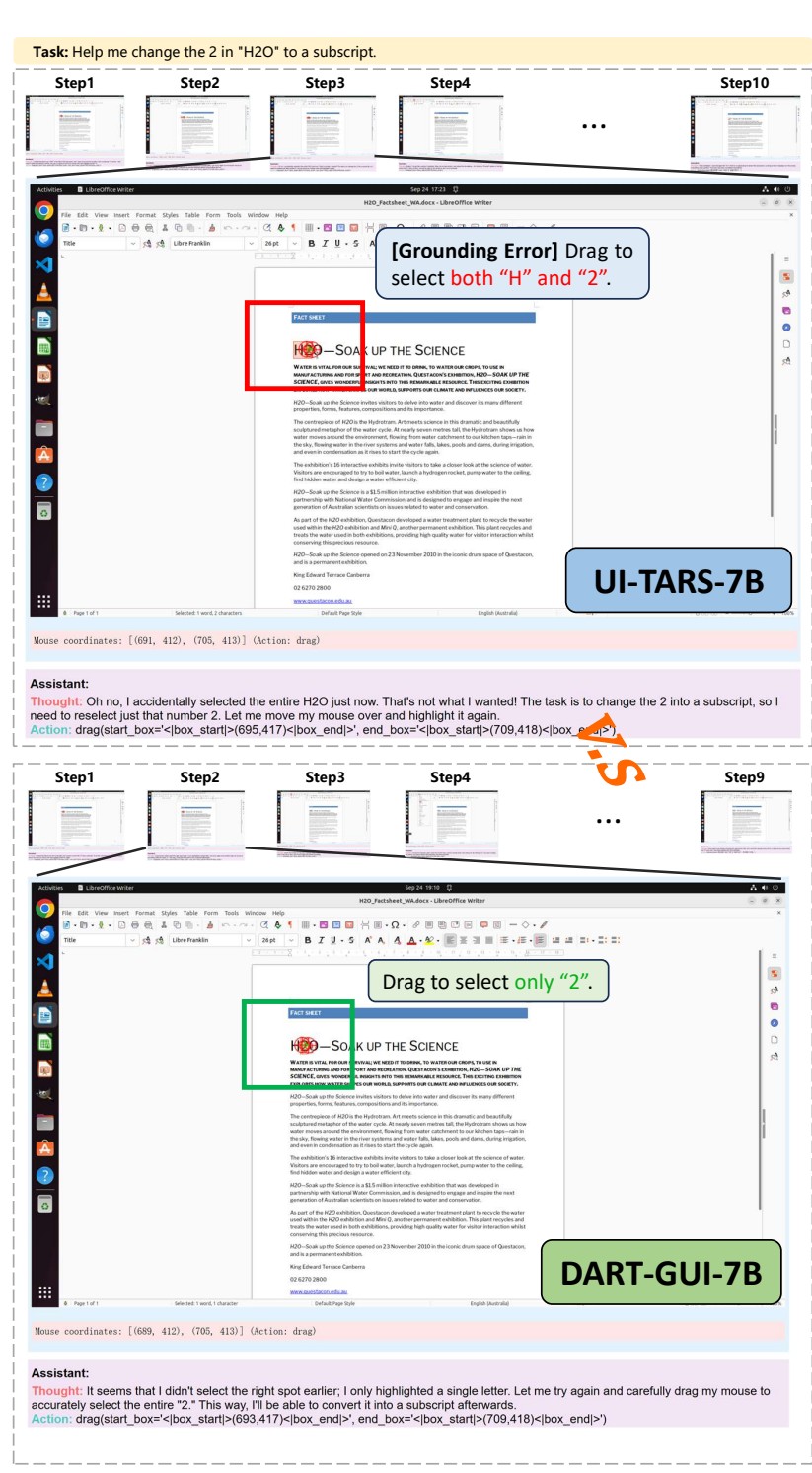

Figure 11: Case study comparing UI-TARS-7B and DART-GUI-7B on editing text in LibreOffice. UI-TARS-7B makes a grounding error by selecting both "H" and "2" in "$H_2O$", whereas DART-GUI-7B correctly highlights only the "2" for conversion into a subscript.

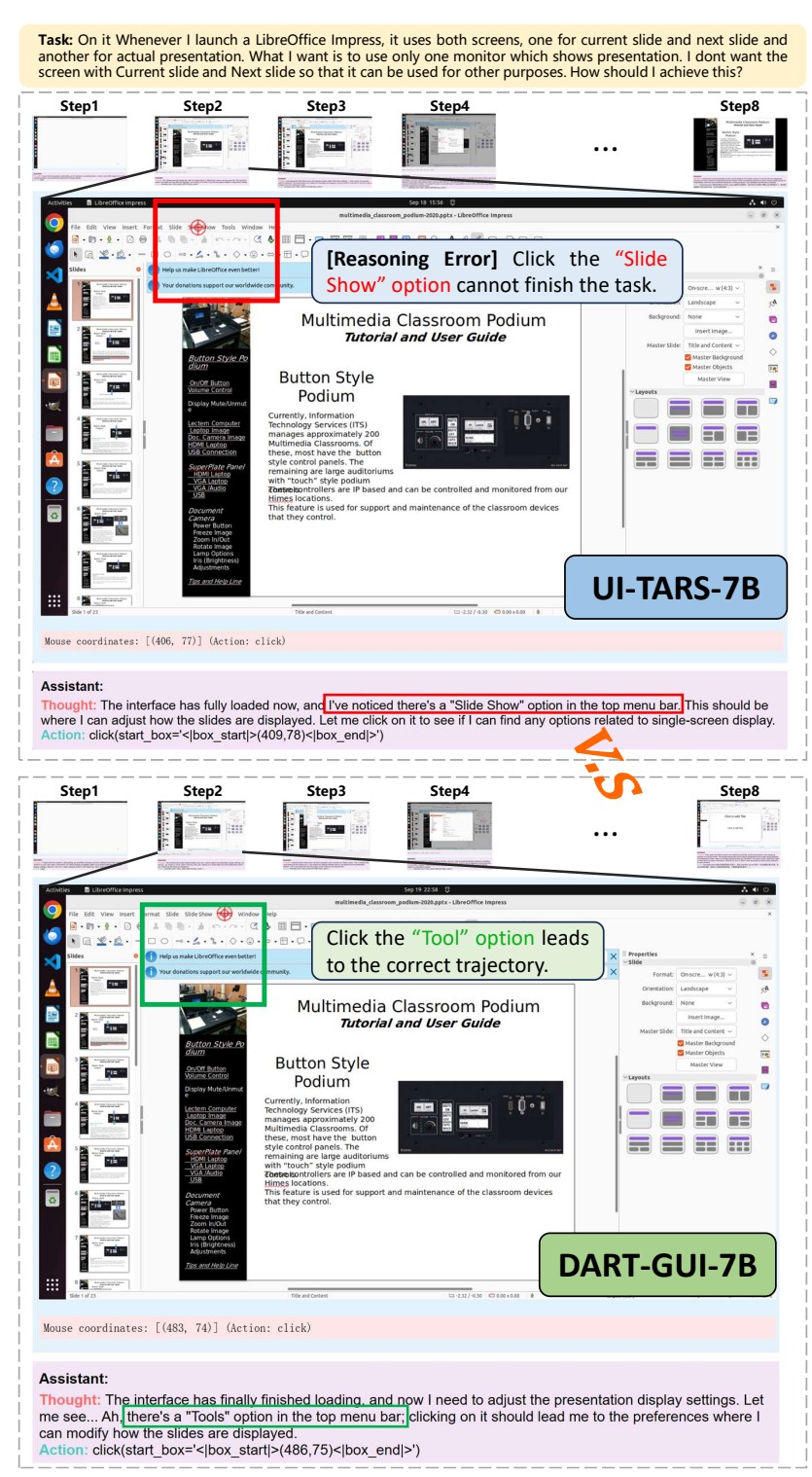

Figure 12: Case study on an extremely difficult LibreOffice Impress task. The task requires configuring dual-slide display settings. The baseline model (top) incorrectly clicks "Slide Show" in the menu, leading to task failure. Our DART-GUI-7B model (bottom), trained with successful trajectories from the trajectory pool, correctly selects "Tools" to access the preferences panel where dual-slide display can be configured. This demonstrates our model's ability to learn from rare successful trajectories and solve previously intractable tasks through RL training.

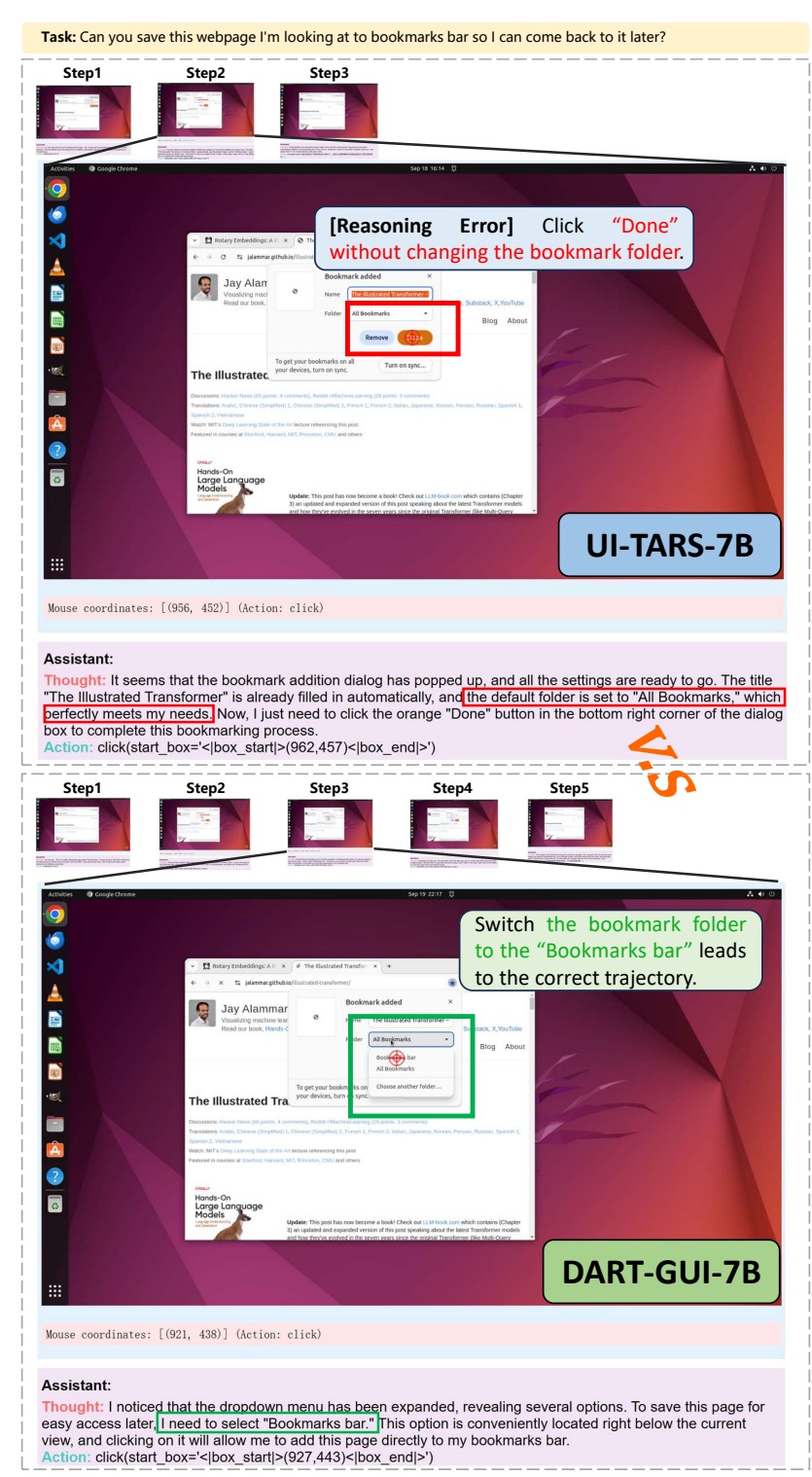

Figure 13: Case study on an extremely difficult bookmark saving task. The task requires saving a webpage to the bookmarks bar for quick access. The baseline model (top) makes a critical error by clicking "Done" without changing the bookmark folder from the default "All Bookmarks" to "Bookmarks bar," resulting in task failure. Our DART-GUI-7B model (bottom) correctly identifies the need to switch the folder dropdown to "Bookmarks bar" before confirming, successfully completing the task. This demonstrates our model's ability to understand subtle but crucial UI requirements that determine task success, learned through RL training on rare successful trajectories.

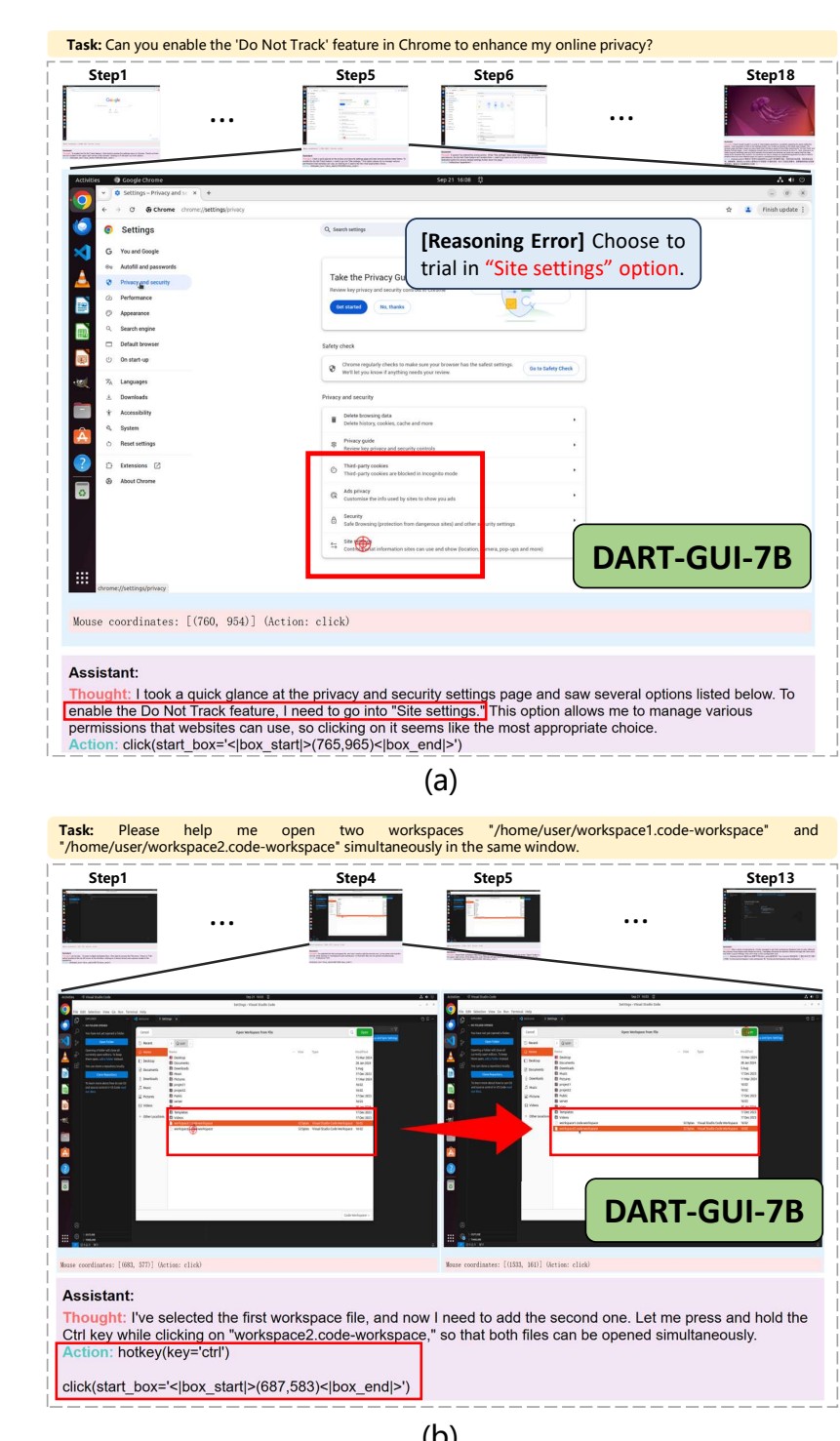

(a)

(b)

Figure 14: Failure cases of DART-GUI-7B. (a) For the task of enabling the "Do Not Track" feature in Chrome, the model incorrectly clicks the "Site settings" option instead of the "Third-party cookies" option required to access the relevant privacy control. (b) For the task of opening two workspaces simultaneously in VS Code, the model attempts a Ctrl+click sequence, but due to action space limitations, the execution corresponds to sequentially pressing "Ctrl" and clicking without holding "Ctrl", which deselects the first workspace and leaves only the second one selected.

remaining errors) and Grounding Errors (16.4% to 7.7%). As the model becomes more capable of handling reasoning and grounding challenges, Knowledge Limitations (rising to 38.5%) emerge as the primary bottleneck for the remaining unsolved tasks.

We analyze the correlation between failure modes and trajectory length. As shown in Figure 16, successful trajectories from DART-GUI-7B exhibit a remarkably even distribution across all length intervals (consistently ranging between 15% and 19%), demonstrating robustness across both short and long-horizon tasks. In contrast, baseline failures are heavily skewed toward the maximum length (peaking at 31.2%), indicating that the baseline model often struggles to terminate effectively or enters stuck states in complex scenarios.

Figure 17 presents the Pearson correlation coefficients between specific failure modes and task metrics. The statistical analysis identifies key failure patterns. There is a notable positive correlation (0.29) between Looping/Stuck states and trajectory length, confirming that agents stuck in repetitive loops tend to exhaust the step limit. Additionally, Knowledge Limitations show a positive correlation (0.18) with task complexity, suggesting that as tasks become more intricate, the primary challenge shifts from execution to a lack of specific domain knowledge.

Finally, we observe that failure modes vary significantly depending on the application interface, as illustrated in Figure 18. LibreOffice Calc, characterized by a dense grid of small cells, is dominated by Grounding Errors (100%), reflecting the difficulty in precise coordinate prediction. In contrast, applications like Chrome and LibreOffice Writer, which involve multi-step menu navigation, primarily exhibit Reasoning Failures (>75%), indicating that the main challenge in these environments lies in logical planning rather than visual element identification.

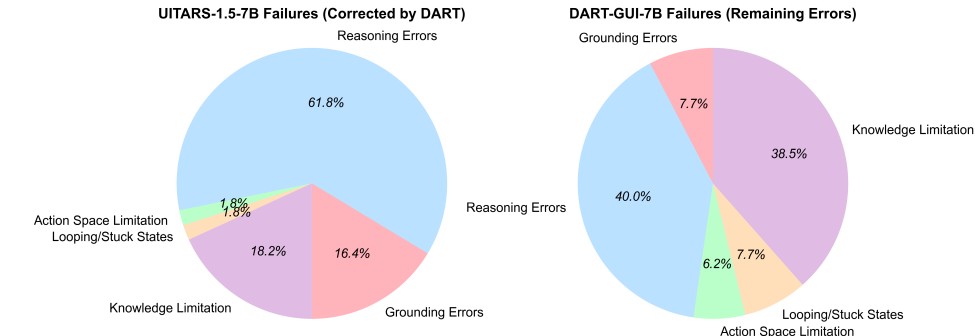

Figure 15: Comparison of Failure Mode Distributions. The charts contrast the error distribution of tasks failed by the baseline but corrected by DART (Left) against the remaining failures of DART-GUI-7B (Right).

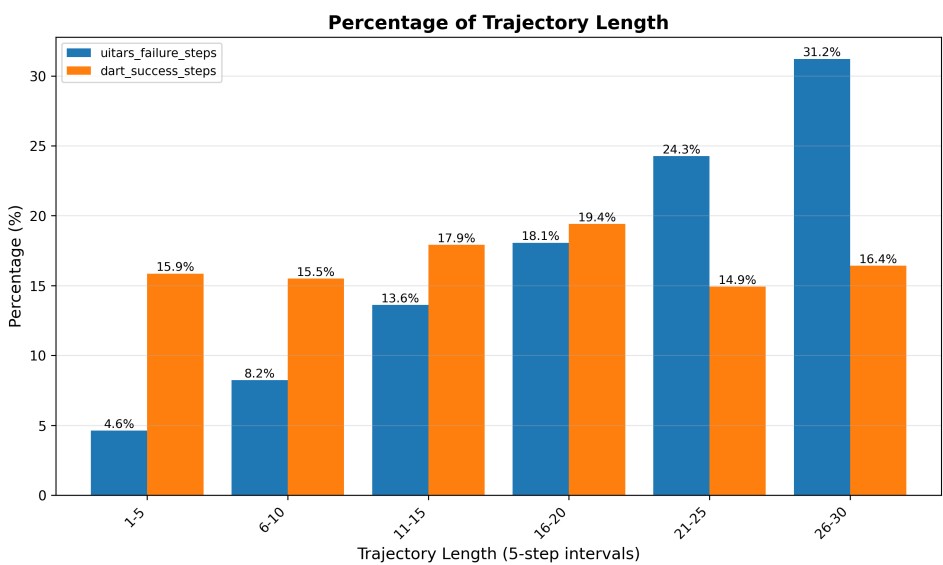

Figure 16: Distribution of Trajectory Lengths. A comparison between the trajectory lengths of DART's successful tasks (Orange) and the baseline's failed tasks (Blue).

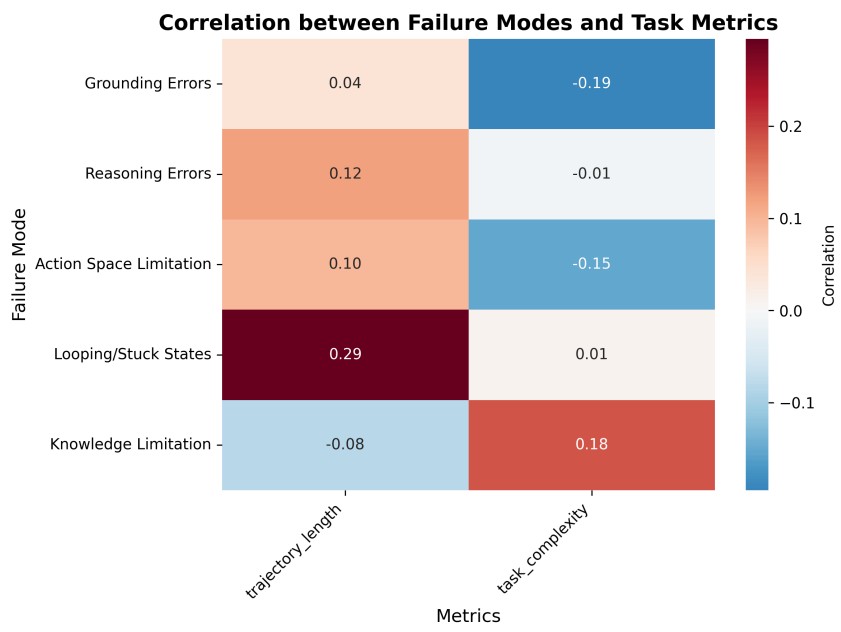

Figure 17: Correlation Matrix of Failure Modes and Task Metrics. Heatmap displaying the Pearson correlation coefficients between specific failure modes and task metrics (Trajectory Length and Task Complexity).

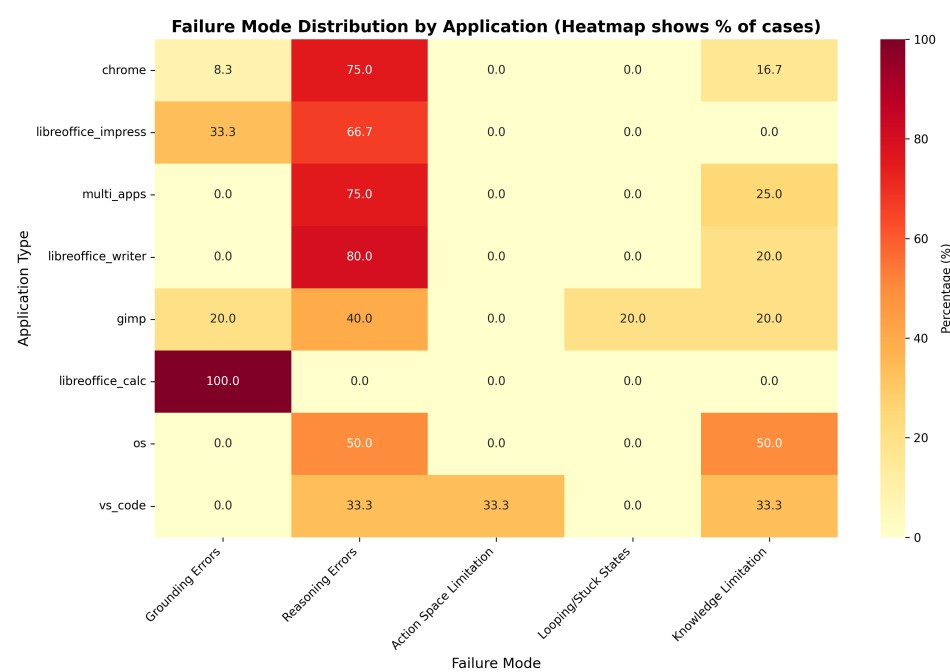

Figure 18: Failure Mode Distribution by Application Type. Heatmap showing the percentage of failure modes across different GUI applications.

