# OpenReview forum: "Efficient Multi-turn RL for GUI Agents via Decoupled Training and Adaptive Data Curation"
_ICLR.cc/2026/Conference — Submitted to ICLR 2026_

### Official Review · Reviewer_xFpq · 2025-10-22

**Soundness:** 3
**Presentation:** 4
**Contribution:** 3
**Rating:** 6
**Confidence:** 3

**Summary:**

This paper introduces DART, a decoupled RL framework for GUI agents that splits training into four async parts (environment cluster, rollout service, data manager, trainer) to boost throughput and hardware use. It also adds adaptive data curation: pre-collecting successful trajectories for hard tasks, adjusting rollout count and max length by task difficulty, training on high-entropy steps, and stabilizing updates with truncated importance sampling. On OSWorld, the 7B model trained with DART reaches 42.13% success, a 14.61% absolute gain over the base and above prior open-source results.

**Strengths:**

1. The paper splits RL for GUI agents into four async parts so training and interaction never block each other, which is a clear design shift from coupled systems. This decoupling is novel for this setting and is backed by analysis on different parts.

2. The method adds adaptive data curation at several levels (task, trajectory, step, token), including high-entropy step focus and truncated importance sampling to stabilize updates. The system describes concrete components and pipelines (Trainer, Rollout Service, Env Cluster, Data Manager) with practical details. It also documents system prompts, rollout settings, hardware, and database tables that track trajectories and checkpoints.

3. The approach improves utilization and throughput (up to 1.6× GPU for rollout, 1.9× training throughput, 5.5× env utilization). It also lifts OSWorld success to 42.13%, a +14.61 point gain over the base and above prior open-source results.

4. The authors plan to open-source the framework, model checkpoints, and curated data, which can speed up progress on GUI agents. This combined with detailed system reporting, makes the work quite useful.

**Weaknesses:**

1. Training and evaluation are both on OSWorld, and the training pool is a 203-task subset from that same benchmark. This raises the risk of benchmark-specific tuning / overfitting; adding at least one other GUI suite or an OOD test would better show robustness.

2. Results compare models under different step budgets, which can blur sample-efficiency claims. The system also relies on substantial infra (~180 parallel envs), so adding cost-normalized metrics would clarify practical efficiency.

3. Ablations would be better with stronger controls. The ablation studies probe DR/DTL/HE/DA, but only on 45 training tasks. Expanding these to the full split or the test set would be more convincing. Also report “coupled + curation” vs. “decoupled + curation” to isolate the framework effect on final OSWorld scores.

4. Pre-collecting successful trajectories for hard tasks may nudge behavior toward memorizing specific solutions. Please add controls where such pools are withheld, and report how performance changes. The method also hinges on truncated importance sampling and a fixed high-entropy threshold (top 80% of steps). Please include sensitivity sweeps over the clipping cap, entropy cut, etc.

**Questions:**

1. Since the training and evaluation both center on OSWorld, can you add any OOD or cross-suite tests to show generalization? Or is there any difficulty doing so?

2. In Figure 1(c), why does the training accuracy fluctuate so significantly? In this case, would selecting a checkpoint at a local maximum result in an unreliable estimate?

---

> ### Author Response · Authors · 2025-11-23
> **Response to Reviewer xFpq [1/3]**
>
> Thank you for the comprehensive and encouraging review. We appreciate your recognition of DART as a novel design shift towards decoupled systems in GUI agent training, as well as the value of the practical system details and significant performance gains on OSWorld we reported. We are also glad you highlighted the potential impact of our open-source plan on speeding up community progress. To simplify the discussion, we use **W** for each weakness, **Q** for each question, and **A** for our response.
>
> > **W1:** Training and evaluation are both on OSWorld, and the training pool is a 203-task subset from that same benchmark. This raises the risk of benchmark-specific tuning / overfitting; adding at least one other GUI suite or an OOD test would better show robustness.
> > **Q1:** Since the training and evaluation both center on OSWorld, can you add any OOD or cross-suite tests to show generalization? Or is there any difficulty doing so?
>
>
> **A.** Thanks for your insightful feedback. To demonstrate that our model has learned **robust interaction capabilities** rather than simply memorizing specific trajectories, we conducted three levels of verification:
>
> **1. Robustness against Environmental Dynamics**
> First, we clarify that "memorization" is inherently ineffective in OSWorld. As noted in [a], the environment is highly stochastic: initial states, window positions, and pop-ups vary dynamically across episodes. Memorizing a fixed action sequence that works in one episode will likely fail in the next. The model must learn to react to visual feedback of dynamic environments rather than overfit to a static sequence.
>
> **2. Performance on Held-out Tasks (Unseen Tasks)**
> To empirically verify that the model has not overfitted the training set, we constructed a **held-out validation set** of 171 new tasks, distinct from the 203 training tasks. As shown in Table G, DART-GUI-7B achieves **36.06%**, significantly outperforming the baseline. This confirms the policy remains effective on unseen user intents within the same domain.
>
> **Table G:** Comparison between UI-TARS-1.5-7B (baseline) and ours on the 171 human-annotated held-out tasks.
> |Models|Acc (%) on Held-out Tasks (N=171) |
> | :--- | :--- |
> | UI-TARS-1.5-7B | 27.19 ± 2.14 |
> | **DART-GUI-7B (Ours)** | **36.06 ± 1.38** |
>
> **3. Performance on External Benchmark (Unseen Domain)**
> To further rule out overfitting to the OSWorld simulator itself, we extended our evaluation to **UI-Vision [b]**, a completely different desktop GUI benchmark. We evaluated models in a strict **zero-shot** setting. As shown below, DART-GUI-7B consistently outperforms the baseline, indicating robustness across different interface distributions.
>
>
> **Table H:**  Zero-shot performance on UI-Vision.
> | Model \ Split | Basic | Functional | Spatial | Overall |
> | :---: | :---: | :---: | :---: | :---: |
> | UI-TARS-1.5-7B | 20.82% | 21.84% | 7.49% | 16.44% |
> | **DART-GUI-7B** | **21.67%** | **22.40%** | **8.48%** | **17.24%** |
>
> [a] OpenCUA: Open Foundations for Computer-Use Agents.
>
> [b] UI-Vision: A Desktop-centric GUI Benchmark for Visual Perception and Interaction.
>
>
> > **W2:** Results compare models under different step budgets, which can blur sample-efficiency claims. The system also relies on substantial infra (~180 parallel envs), so adding cost-normalized metrics would clarify practical efficiency.
>
>
> **A.** We clarify that there is a misunderstanding regarding the metrics. The "Max Steps" in Table 1 refers to the **inference-time constraint** (maximum steps allowed during evaluation), not the training sample budget. Our sample-efficiency claim in line 371 refers to **inference action efficiency**, specifically, the agent's ability to complete tasks using significantly fewer steps. Achieving a higher success rate with a stricter limit (30 steps) compared to baselines (100 steps) demonstrates superior decision-making capabilities.
>
> To ensure a strictly fair comparison, we re-evaluated the baselines under the same **30-step limit**, whose max-step is less than 30 in the original manuscript. As shown below, DART maintains a significant lead under identical budgets:
>
> **Table I:** Comparision on the fullset of OSWorld with max step=30.
> | Model | Max Steps | Task Success Rate |
> | :--- | :---: | :---: |
> | UI-TARS-1.5-7B | 30 | 24.17% |
> | GUI-Owl-7B | 30 | 34.79% |
> | **DART-GUI-7B (Ours)** | **30** | **42.13%** |
>
> We have revised the manuscript to replace "sample efficiency" with "inference efficiency" to prevent future misunderstanding.

---

> > ### Author Response · Authors · 2025-11-23
> > **Response to Reviewer xFpq [2/3]**
> >
> > > **W3:** Ablations would be better with stronger controls. The ablation studies probe DR/DTL/HE/DA, but only on 45 training tasks. Expanding these to the full split or the test set would be more convincing. Also report “coupled + curation” vs. “decoupled + curation” to isolate the framework effect on final OSWorld scores.
> >
> >
> > **A.** We sincerely thank the reviewer for this constructive suggestion. We agree that full-set ablations provide stronger evidence for the method's robustness.
> >
> > **1. Validity of the 45-Task Subset**
> > Regarding the **45-task subset** used in our original ablation studies, we clarify that this was not a random selection. To ensure statistical representativeness under computational constraints, we employed **stratified sampling** based on the baseline model's difficulty tiers. Specifically, we selected 15 tasks from each of three difficulty levels (Pass@32: 0-30%, 30-60%, and 60-80%). This design ensures that the subset mirrors the difficulty distribution of the full benchmark, making the observed ablation trends reliable indicators of overall performance.
> >
> > **2. Full-Set Ablation Analysis**
> > While computational constraints during the rebuttal prevented us from completing full-scale training for every variant, we prioritized the core components to isolate the framework's effect. We conducted experiments on the **full OSWorld benchmark** to verify the impact of the Decoupled Framework itself and the Distribution Alignment (DA) module. The results are presented in **Table J**.
> >
> > **Table J:** Ablation study on the full OSWorld benchmark (Max Steps=30).
> > | Setting | Task Success Rate | Improvement |
> > | :--- | :---: | :---: |
> > | UI-TARS-1.5-7B | 24.17% | - |
> > | Ours (Decoupled w/o DA) | 34.77% | +10.60% |
> > | Ours (Decoupled w/ DA) | 38.81% | +4.04% |
> > | **Ours (Decoupled w/ DR+DTL+HE+DA)** | **42.13%** | **+3.32%** |
> >
> > *Note: "Ours (Full Method)" includes DA + Dynamic Rollout (DR) + Dynamic Trajectory Length (DTL) + High Entropy (HE).*
> >
> > **Analysis of Framework Effects:**
> > *   **Effect of Decoupling (Framework Gain):** Even without the Distribution Alignment module (`Ours w/o DA`), conducting RL using our decoupled architecture achieves **34.77%**, significantly outperforming the baseline (**24.17%**). This isolates the benefit of the framework itself: the asynchronous design allows for vastly more efficient data collection and training throughput.
> > *   **Effect of Distribution Alignment (DA):** Adding the DA module (`Ours w/ DA`) boosts performance further to **38.81%**. This confirms that correcting the distributional shift between the rollout and training policies is critical for stability at scale.
> > *   **Full Optimization:** The complete method (`Full Method`), incorporating DR, DTL, and HE, yields the final performance of **42.13%**.
> >
> > We will include the complete full-set ablations for Dynamic Rollout (DR), Dynamic Trajectory Length (DTL), High Entropy (HE), and the "Coupled vs. Decoupled" comparison in the next version of the manuscript.

---

> > > ### Author Response · Authors · 2025-11-23
> > > **Response to Reviewer xFpq [3/3]**
> > >
> > > > **W4:** Pre-collecting successful trajectories for hard tasks may nudge behavior toward memorizing specific solutions. Please add controls where such pools are withheld, and report how performance changes. The method also hinges on truncated importance sampling and a fixed high-entropy threshold (top 80% of steps). Please include sensitivity sweeps over the clipping cap, entropy cut, etc.
> > >
> > > **A.** We appreciate the reviewer's detailed scrutiny regarding data handling and hyperparameter selection.
> > >
> > > **1. Impact of Pre-collected Trajectories & Memorization Risk**
> > >
> > > We clarify that the Experience Pool is primarily a cost-saving mechanism to mitigate the high sampling cost of difficult tasks, rather than a driver for memorization. Each trajectory (including pre-collected ones) is used strictly **once** during training.
> > >
> > > To empirically prove that our performance does not stem from memorizing these specific trajectories, we provide two pieces of evidence:
> > >
> > > *   **Ablation Study (Table K):** We retrained the model without any pre-collected trajectories. As shown below, the performance drop is marginal (**1.11%**), demonstrating that the model can successfully learn robust policies through online sampling alone, albeit with higher computational cost.
> > > *
> > >     **Table K:** Ablations without pre-collected trajectories.
> > >     | Configuration | Success Rate (%) |
> > >     | :--- | :---: |
> > >     | DART w/o pre-collected trajectories | 41.02 |
> > >     | **DART (Full Method)** | **42.13** |
> > >
> > > *   **SFT Control Experiment (Table L):** To further rule out trajectory-level memorization, we trained a standard SFT model on **34k successful steps** (covering 181 tasks) from the base model. In the dynamic OSWorld environment, memorization fails: SFT achieves only **28.14%**, significantly underperforming our RL method (**42.13%**). This confirms DART learns robust, reactive policies rather than fixed paths.
> > > *
> > >     **Table L:** Comparison with SFT baseline (Max steps=30).
> > >     | Method | Acc (%) |
> > >     | :--- | :---: |
> > >     | UI-TARS-1.5-7B | 24.17 |
> > >     | UI-TARS-1.5-7B-SFT | 28.14 |
> > >     | **DART-GUI-7B (Ours)** | **42.13** |
> > >
> > > **2. Hyperparameter Selection Strategy**
> > >
> > > Regarding the hyperparameter settings, our selections were guided by established empirical guidelines and specific design motivations to ensure stability within our decoupled framework:
> > >
> > > *   **Truncated Importance Sampling:** We set a conservative upper bound of 1.0 to act as a stabilizer. In our decoupled setting, the rollout model (quantized) inherently differs from the training model. Capping the weight at 1.0 ensures that we only penalize distribution shifts (where the rollout policy deviates) without artificially boosting updates, effectively mitigating variance caused by quantization noise.
> > > *   **High-Entropy Threshold:** GUI tasks often involve many trivial, low-entropy steps (e.g., scrolling, waiting). Following the analysis in **[c]**, we set the threshold to 0.8 to filter out these non-essential tokens, forcing the RL algorithm to focus credit assignment on the "critical decision points" that actually determine task success.
> > > *   **Clipping Bounds:** We adopted the dynamic clipping configuration from **DAPO [d]**, which has been empirically validated for stabilizing large-scale LLM reinforcement learning.
> > >
> > > Due to computational resource constraints, we were unable to complete extensive sensitivity sweeps for these parameters during the rebuttal period. We commit to conducting these analyses and including them in the next version.
> > >
> > > [c] Beyond the 80/20 Rule: High-Entropy Minority Tokens Drive Effective Reinforcement Learning for LLM Reasoning.
> > >
> > > [d] DAPO: An Open-Source LLM Reinforcement Learning System at Scale.
> > >
> > >
> > > > **Q2:** In Figure 1(c), why does the training accuracy fluctuate so significantly? In this case, would selecting a checkpoint at a local maximum result in an unreliable estimate?
> > >
> > > **A.** In Figure 1(c), the curve plots the success rate of the **sampled tasks in the current step**, not the full dataset. Validating on all 203 tasks at every step is computationally prohibitive. Consequently, the reason for the curve's fluctuation is that different steps are evaluated on different task numbers/difficulties.
> > >
> > >
> > > Regarding reliability, we did not cherry-pick a local maximum. We selected the **final checkpoint** (step 80) for evaluation, ensuring the reported performance reflects the model's final converged state.

---

### Official Review · Reviewer_NshA · 2025-10-22

**Soundness:** 2
**Presentation:** 2
**Contribution:** 2
**Rating:** 4
**Confidence:** 4

**Summary:**

This paper addresses the bottlenecks hindering the application of reinforcement learning (RL) to Vision-Language Model (VLM) based GUI agents: the inefficiency of training systems and the lack of high-quality interaction data. The paper introduces DART, a Decoupled Agentic RL Training framework, which presents a comprehensive solution. Architecturally, DART deconstructs the RL loop into four asynchronous, non-blocking modules, a design that achieve 5.5x increase in environment utilization and a 1.9x boost in training throughput.

Algorithmically, this efficient data generation engine selects the most informative experiences for learning. Effectively solve challenges like sparse rewards and imbalanced learning. The resulting model, DART-GUI-7B, achieves a 42.13% success rate on the OSWorld benchmark  by solving tasks in fewer steps than its competitors. The work's commitment to fully open-sourcing the framework, data, and model further solidifies its contribution to the field.

**Strengths:**

1. The strength of this work lies in its sophisticated systems engineering with algorithmic design.
2. The DART framework is a engineering achievement. by decoupling the training process, solving the system-level bottleneck of slow environment interaction, a claim substantiated by the transformative 5.5x improvement in environment utilization.
3. High-entropy-driven step selection and Experience Pool for difficult tasks demonstrate a understanding of the learning dynamics in GUI environments.

**Weaknesses:**

1. Although the author claims the SOTA performance on open source model on OS-world. If look at all ranking in os-world, the top8 performance already can achieve over 40% performance. The top1 UI-TARS-2-2509 already achieve 53.1% performance. Even though they are not open source, but I think more discussion and analysis between proposed model and them will help.
2. For experimence pool, the paper mentions "pre-populate the Experience Pool by collecting and storing high-quality successful trajecctories by preliminary sampling". It will be good to explain more on how trajectories are generated, which model are used when generated the trajectory. The discussion on the computation cost on preliminary sampling will be good. More details on each RL training step's successful trajectory rate will help.
3. For framework effiecient analysis, more details on baseline experiment setting will help. Current academic and industry also have lots of work on multiturn RL framework. More performance comparison between the related work like sky-rl will help. In figure 4, it shows the performance comparision on 4 GPUs and 80 Environment, larger scale experiments setting is needed to enhance paper's performance statement.

**Questions:**

Please check weaknesses.

---

> ### Author Response · Authors · 2025-11-23
> **Response to  Reviewer NshA [1/2]**
>
> Thank you for the insightful review. We appreciate your recognition of DART’s combination of sophisticated systems engineering with algorithmic design, particularly the transformative 5.5x improvement in env util. To simplify the discussion, we use **W** for weakness, and **A** for our response.
>
> > **W1:** Although the author claims the SOTA performance on open source model on OS-world. If look at all ranking in os-world, the top8 performance already can achieve over 40% performance. The top1 UI-TARS-2-2509 already achieve 53.1% performance. Even though they are not open source, but I think more discussion and analysis between proposed model and them will help.
>
> **A.** Thank you for your valuable feedback. We would like to clarify that UI-TARS-2-2509 was released subsequent to our paper's submission deadline. We recognize UI-TARS-2 and other close-source models as significant, large-scale systems that achieve its impressive results by integrating extensive data, resources, larger model size (230B for UI-TARS-2), and multiple training phases. In contrast, DART focuses on the efficiency and effectiveness of the RL training methodology specifically for accessible 7B open-source models. Therefore, we position our contribution as an orthogonal and complementary methodology focused on a crucial part of the training pipeline, rather than a direct competitor to the overall performance of such a large scale GUI system. We have updated the Table 1 in the revised version with the new results on the OSWorld benchmark for a more comprehensive comparison.
>
> > **W2:** For experimence pool, the paper mentions "pre-populate the Experience Pool by collecting and storing high-quality successful trajecctories by preliminary sampling".
>
> > **W2.1:** It will be good to explain more on how trajectories are generated, which model are used when generated the trajectory.
>
> **A.** The trajectories were generated using our baseline model, *UI-TARS-1.5-7B*. We performed preliminary sampling on the training set with a rollout of $N=32$. Tasks with a success rate lower than 1/8 were identified as "hard tasks," and their successful trajectories were stored in the Experience Pool. During online training (where rollout $N=8$), if the agent fails to generate a correct trajectory for these hard tasks, we retrieve a successful one from this pre-collected pool to guide the update. All the trajectories are used only once, even in the experience pool.
>
> > **W2.2:** The discussion on the computation cost on preliminary sampling will be good.
>
> **A.** The preliminary sampling utilized **8 $\times$ H100 GPUs** paired with **80 $\times$ Ubuntu Docker containers** (allocated 3 CPU cores each). The maximum trajectory length was set to 100 steps, and the total data collection process took approximately **48 hours**.
>
> > **W2.3:** More details on each RL training step's successful trajectory rate will help.
>
> **A.** Figure 1( c ) illustrates the relationship between the "successful trajectory rate" and RL training steps. Each data point in the figure represents the average success rate of the specific batch of tasks sampled at that current step, rather than the success rate across all 203 training tasks.
>
> > **W3.1:** For framework effiecient analysis, more details on baseline experiment setting will help.
>
> **A.** To ensure a rigorous comparison, our non-decoupled baseline implements a standard synchronous RL paradigm (following ARPO [a]), where the workflow proceeds sequentially through 'Sampling → Training → Model Syncing → Sampling' loops. In this setup, each rollout worker is bound to specific environment instances, while the global training step blocks progress until the entire batch is completed by all workers. Crucially, to isolate architectural efficiency from hardware variables, we maintained strictly identical resource allocations for both settings: utilizing 80 environments and approximately 300 NVIDIA H100-80G GPU-hours over 100 training steps.
>
> We quantified the efficiency using the following metrics:
>
> *   **Environment Utilization ($U_{\text{env}}$):** The ratio of time environments spend actively executing actions versus the total available time.
>     $$U_{\text{env}} = \frac{\sum_{i=1}^{N_{\text{env}}} t_{\text{active}}^{(i)}}{T_{\text{total}} \times N_{\text{env}}} \times 100\%$$
>     where $t_{\text{active}}^{(i)}$ denotes the active execution time of the $i$-th environment, $N_{\text{env}}$ is the total number of environment instances (80), and $T_{\text{total}}$ is the total wall-clock time of the experiment.
>
> *   **Training Throughput ($T_{\text{throughput}}$):** The average number of action steps collected and processed per minute.
>     $$T_{\text{throughput}} = \frac{N_{\text{actions}}}{T_{\text{total}} \text{ (min)}}$$
>     where $N_{\text{actions}}$ represents the total number of valid action steps used in the training period.
>
> We have added these implementation details and definitions to the appendix to clarify the efficiency analysis.

---

> ### Author Response · Authors · 2025-11-23
> **Response to Reviewer NshA [2/2]**
>
> > **W3.2:** Current academic and industry also have lots of work on multiturn RL framework. More performance comparison between the related work like sky-rl will help.
>
>
> **A.** We appreciate the reviewer for highlighting Sky-RL[a], which is an excellent framework for multi-turn RL. However, Sky-RL is primarily tailored for text-based tool use and API calling scenarios (e.g., SWE-bench). To address the performance comparison conceptually, we analyze how DART and Sky-RL tackle the distinct efficiency comparison inherent to their respective domains in Table E. We have incorporated a brief discussion comparing to Sky-RL in the **Related Work** section and added **Table E** with an analysis in **Appendix** of our revision.
>
> **Table E:** Comparison of efficiency bottlenecks and architectural optimization strategies between DART and Sky-RL.
>
> | Feature | **DART (Ours)** | **Sky-RL** |
> | :--- | :--- | :--- |
> | **Target Domain** | **Desktop GUI** (Visual-heavy) | **Code/Research** (Text/Logic-heavy) |
> | **Primary Bottleneck** | **VLM Inference & Rendering** (Encoding high-res screenshots) | **Tool Execution Latency** (CPU-bound tasks like running tests) |
> | **Efficiency Strategy** | **Decoupled & Asynchronous** (Separating Trainer/Rollout to maximize VLM throughput) | **Async Pipeline** (Hiding CPU tool latency behind GPU inference) |
> | **Model Sync** | **Per-Worker Non-Blocking Sync** (Staggered updates to prevent idle time) | **Standard/Pipelined Sync** (Optimized for data flow consistency) |es to prevent idle time) | **Standard/Pipelined Sync** (Optimized for data flow consistency) |
>
>
> For the empirical performance comparison, we focus on frameworks natively designed for GUI agents: ARPO[b] and Zero-GUI[c]. The results are shown in Table F.
>
> **Table F:** Estimated Efficiency and Performance Comparison.
> | Framework | Architecture | Reward Mechanism |  Training Throughput |Env Util |  Success Rate |
> | :--- | :--- | :--- | :---: | :---: |:--:|
> | **ARPO [b]** | Coupled (Sync) | Rule-based | 22.6 actions/min | 12.2% | 29.9% |
> | **Zero-GUI [c]** | Coupled (Sync) | VLM-based (4x Voting) | 19.4 actions/min| 11.4% | 20.2% |
> | **DART (Ours)** | **Decoupled (Async)** | **Rule-based** | **43.6 actions/min** |**67.7%** |**42.13%** |
>
>
>
> **Analysis:**
> 1.  **vs. ARPO:** ARPO employs a standard synchronous rollout-training loop, architecturally equivalent to our **"Non-Decoupled" baseline**. Thus, its throughput is limited (~22.6 actions/min). DART achieves a **1.9x speedup** and significantly higher accuracy (**42.13%** vs 29.9%).
> 2.  **vs. Zero-GUI:** Zero-GUI is strictly slower than the ARPO baseline. It introduces substantial overhead by replacing rule-based verification with an LLM-based reward model, requiring **4x VLM inferences per trajectory**. DART achieves superior performance with a lightweight rule-based reward system.
>
>
> > **W3.3:** In figure 4, it shows the performance comparision on 4 GPUs and 80 Environment, larger scale experiments setting is needed to enhance paper's performance statement.
>
>
> **A.** We thank the reviewer for this insightful comment. We would like to clarify that Figure 4 was designed as a **simplified schematic** to visually explain the mechanism of per-worker synchronization, rather than representing the actual scale of our experiments.
>
> In fact, our main experiments were conducted on a significantly larger scale, utilizing a cluster of *8$\times$NVIDIA H100 GPUs* to orchestrate *180 parallel Docker environment*s. This setup represents a high workload density (*~22.5 environments per GPU*), which we believe effectively validates the system's scalability and its ability to handle massive parallel rollouts without blocking. We have updated the caption in Figure 4 to explicitly state that it is a schematic visualization, ensuring the actual experimental scale is clearly communicated to future readers.
>
> [a] SkyRL-Agent: Efficient RL Training for Multi-turn LLM Agent.
>
> [b] ARPO: End-to-End Policy Optimization for GUI Agents with Experience Replay.
>
> [c] ZeroGUI: Automating Online GUI Learning at Zero Human Cost.

---

### Official Review · Reviewer_WyKM · 2025-10-29

**Soundness:** 3
**Presentation:** 4
**Contribution:** 2
**Rating:** 6
**Confidence:** 4

**Summary:**

This paper introduces DART, a reinforcement learning framework designed to train vision-language model (VLM) based GUI agents more efficiently. The framework addresses two main challenges: (1) slow multi-turn interactions with GUI environments during policy rollout, and (2) insufficient high-quality agent-environment interactions for policy learning. DART separates the training system into four asynchronous modules (environment cluster, rollout service, data manager, and trainer) and introduces adaptive data curation strategies at multiple granularities (task, trajectory, step, and token levels). The resulting model, DART-GUI-7B, achieves strong results w.r.t. the current SOTA on OSWorld.

**Strengths:**

1. Strong engineering contribution: The decoupled architecture with four asynchronous modules (trainer, data manager, env cluster, rollout service) is well-designed and addresses real bottlenecks in GUI agent RL training.

2. Comprehensive data curation scheme: The multi-level adaptive strategy (task, trajectory, step, token) is thoughtful and well-motivated. Each component addresses a specific challenge in GUI agent training.

3. Thorough experimental validation: Good ablation studies demonstrating the contribution of each component.

**Weaknesses:**

1. **Limited algorithmic novelty beyond system engineering**: The paper primarily uses standard step-wise GRPO with techniques borrowed from prior work (e.g., high-entropy step selection from Wang et al., 2025b). While the system integration of vLLM, veRL, and K8s is solid, the core contribution is engineering-focused rather than algorithmic innovation. Critically, DistRL (Wang et al., cited in the paper) should be discussed as a highly relevant baseline since it also decouples training from interaction in an asynchronous RL framework. The paper should clarify what specific advantages DART provides over DistRL beyond implementation details.

2. Insufficient treatment of reward hacking and generalization challenges: The paper does not adequately address the fundamental problem of reward hacking in GUI agents, which is currently the main bottleneck for generalization. The discussion of distribution shift is obscured by terminology (OOD tokens, quantization mismatch, GRPO variants) without addressing the core challenge: how does the agent generalize beyond trajectory-level reward gaming to learn robust GUI interaction policies? This is especially critical given that the method trains on only **203** tasks but claims generalization.

3. Inadequate failure mode analysis and limited mechanistic understanding: Section A.6 presents only two cherry-picked failure cases without systematic analysis. You may need to give some details, including:

- What percentage of errors are due to reasoning failures vs. grounding errors vs. action space limitations?
- Which of the proposed methods (dynamic rollout, entropy filtering, etc.) are designed to address which specific failure modes?
- How do these failure modes correlate with task complexity, trajectory length, or application type?

**Questions:**

- See point 3 above

- Also, could you give out some trading off when scaling up the actors or interactive agents and going towards highly async environments, what will happen, and how does your method prevent the risks of unstable training?

- How to measure the robustness and generalization abilities of your models since your training set only includes 203 tasks?

---

> ### Author Response · Authors · 2025-11-23
> **Response to Reviewer WyKM [1/4]**
>
> Thank you for the detailed and constructive review. We appreciate your recognition of the strong engineering contribution behind DART's decoupled architecture. To simplify the discussion, we use **W** for each weakness, **Q** for each question, and **A** for our response.
>
> > **W1.** Limited algorithmic novelty beyond system engineering: The paper primarily uses standard step-wise GRPO with techniques borrowed from prior work (e.g., high-entropy step selection from Wang et al., 2025b). While the system integration of vLLM, veRL, and K8s is solid, the core contribution is engineering-focused rather than algorithmic innovation. Critically, DistRL (Wang et al., cited in the paper) should be discussed as a highly relevant baseline since it also decouples training from interaction in an asynchronous RL framework. The paper should clarify what specific advantages DART provides over DistRL beyond implementation details.
>
> **A.** Thank you for raising this important comparison with DistRL [a] as a highly relevant baseline. We acknowledge that both systems employ asynchronous frameworks with decoupled training. However, the distinction between DART and DistRL is indeed fundamental, driven by the different requirements of desktop GUI versus mobile GUI tasks. These domain differences lead to key innovations in system architecture, sampling strategy, and training algorithm. Below we first provide a detailed comparison between DART and DistRL, followed by an explanation of DART's distinct advantages.
>
> ### Comparison: DART vs. DistRL
> | Aspect | DART (ours) | DistRL |
> | :-- | :-- | :-- |
> | **Domain** | Desktop GUI | Mobile GUI |
> | **Task Time** | 5-20 minutes per episode | Seconds to minutes per episode |
> | **Decoupled Modules** | Four Modules: `Trainer`, `Rollout Service`, `Environment Cluster`, `Data Manager` | Two Modules: `Host Learner`, `Worker` (integrated inference + environment) |
> | **Samples** | High-freshness, single-use trajectories | Replay-based, reused trajectories |
> | **Training Algorithms** | Step-wise GRPO, a nearly **on-policy** RL algorithm | A-RIDE, an **off-policy** RL algorithm with off-policy corrections |
> | **Rewards & Advantages** | Rule-based rewards (OSWorld scripts) + Group-normalized advantages | LLM-based rewards (Gemini-1.5-Pro) + Learned value network |
>
> ### DART's Specific Advantages Beyond Implementation Details
> The comparison highlights how domain-specific requirements—long task duration and resource-intensive environments in desktop GUI automation—drive DART’s architectural designs and algorithmic choices. DART offers 3 conceptual advantages over DistRL:
>
> 1. **Finer-Grained Decoupling for Long-Duration Tasks**
> DART introduces a three-way decomposition by separating `Rollout Service` from `Environment Cluster`. This design directly addresses the bottleneck of environment availability in desktop GUI tasks, where episodes last 5–20 minutes. In contrast, DistRL’s integrated worker model suffices for mobile tasks lasting seconds to minutes. DART’s finer decoupling improves the environment utilization by $5.5×$ and GPU utilization for rollout by $1.6×$, bringing gains that are intrinsic to the modular design rather than engineering optimizations.
>
> 2. **High-Freshness Sampling for Near-On-Policy Learning**
> DART employs a high-freshness, single-use sampling strategy, where each trajectory is used exactly once and the version gap between rollout and training is constrained to one step. This approach approximates on-policy learning, avoiding the complexity of off-policy corrections. DistRL, by contrast, uses a replay buffer and requires techniques like Retrace(λ) and an auxiliary value network to correct for stale data. DART’s sampling strategy thus ensures greater training stability and higher training throughput ($1.9×$) by avoiding the overhead and approximation errors of off-policy learning.
>
> 3. **Domain-Optimized Algorithmic Simplicity**
> DART uses step-wise GRPO with rule-based rewards and group-normalized advantages, tailored to the long-horizon nature of desktop tasks. This contrasts with DistRL’s use of LLM-based rewards (via Gemini-1.5-Pro) and a learned value network, which introduce additional complexity and potential bias. DART’s choice reflects a principled emphasis on simplicity and reliability: the system handles scalability through architectural parallelism, while the algorithm remains lightweight and transparent.
>
> In summary, DART rethinks asynchronous RL system design for the desktop GUI domain. Its three-level decoupling, high-freshness dataflow, and simplified algorithmic setup are not incremental improvements but foundational adaptations to support efficient and stable on-policy RL for long-duration tasks. These advantages are conceptual in nature and directly address the scalability and stability challenges inherent in desktop GUI automation.
>
> [a] DistRL: An Asynchronous Distributed Reinforcement Learning Framework for On-Device Control Agents.

---

> ### Author Response · Authors · 2025-11-23
> **Response to Reviewer WyKM [2/4]**
>
> > **W2.** Insufficient treatment of reward hacking and generalization challenges: The paper does not adequately address the fundamental problem of reward hacking in GUI agents, which is currently the main bottleneck for generalization. The discussion of distribution shift is obscured by terminology (OOD tokens, quantization mismatch, GRPO variants) without addressing the core challenge: how does the agent generalize beyond trajectory-level reward gaming to learn robust GUI interaction policies? This is especially critical given that the method trains on only 203 tasks but claims generalization.
>
> > **Q2:** How to measure the robustness and generalization abilities of your models since your training set only includes 203 tasks?
>
>
> **A.** Thank you for your insightful comments. We believe that reward hacking is effectively mitigated in our setting due to two intrinsic properties of the OSWorld environment, which make trajectory-level overfitting infeasible:
>
> 1.  **Dynamic Environments:** Unlike static games, desktop GUI environments are inherently stochastic [b]. Initial OS states, window positions, and system pop-ups vary dynamically across episodes. Consequently, a memorized action sequence (trajectory-level gaming) that works in one episode will likely fail due to layout shifts in the next.
> 2.  **Rule-based Verification:** We utilize rule-based evaluation tied to concrete computer-state or file-state changes rather than rewarding specific action paths. To receive a reward, the agent must genuinely complete the task logic; there are no "shortcuts" to trigger a success signal without achieving the user's intent.
>
> To empirically verify that our model is not simply memorizing trajectories, we added three additional experiments.
>
> 1. We conducted a control experiment by training a standard SFT model on **34k successful steps** collected from the base model UI-TARS-1.5-7B. As shown in Table A, while SFT brings a marginal improvement (**28.14%** vs. **24.17%**), it significantly underperforms our RL method (**42.13%**). This performance gap confirms that our RL gains stem from learning robust interaction policies capable of handling dynamic variations, rather than overfitting to fixed trajectories.
>
>
> **Table A.** Comparison between UI-TARS-1.5-7B (baseline), UI-TARS-1.5-7B-SFT (SFT on 34K steps from correct trajectories), and DART-GUI-7B (ours). All models are evaluated on the full set of OSWorld, with max steps set to 30.
>
> | Method  | Acc (%) |
> | :- | :-:|
> | **UI-TARS-1.5-7B**    |   24.17 |
> | **UI-TARS-1.5-7B-SFT** |  28.14 |
> | **DART-GUI-7B (ours)** | **42.13** |
>
> 2. To further rule out overfitting to specific training tasks, we evaluated our model on **a held-out validation set**. We constructed 171 new tasks (distinct from the 203 training tasks) annotated by senior AI researchers, complete with executable evaluation scripts. We tested both the baseline and our model 3 times on this unseen set. As shown in Table B, our method achieves 36.06 ± 1.38%, significantly outperforming the baseline’s 27.19 ± 2.14%. This consistent performance on unseen tasks provides strong evidence that our policy has learned robust GUI interaction patterns rather than memorizing specific task solutions. We will open-source this validation set to facilitate future research.
>
>
>
> **Table B.** Comparison between UI-TARS-1.5-7B (baseline) and ours on the 171 human-annotated held-out tasks.
>
> | Method    | Acc (%) |
> | :-| :-: |
> | **UI-TARS-1.5-7B**    | 27.19 ± 2.14     |
> | **DART-GUI-7B (Ours)** | **36.06 ± 1.38** |
>
>
> 3. Furthermore, to verify that our policy does not overfit to the OSWorld simulator, we extended our evaluation to a widely recognized external benchmark: UI-Vision[c]. These datasets introduce distinct interface distributions and task semantics unseen during training. We evaluated both models in a strict zero-shot setting without fine-tuning. As shown in Table C, DART-GUI-7B consistently outperforms the baseline on both benchmarks. These results confirm that our method captures fundamental GUI interaction capabilities that remain robust across substantial domain shifts, rather than relying on simulator-specific shortcuts.
>
> **Table C.** Zero-shot performance on UI-Vision.
> | Models\Split  | Basic | Functional | Spatical |Overall|
> |:-|:-:|:-:|:-:|:-:|
> | UI-TARS-1.5-7B|20.82% | 21.84% | 7.49%|16.44%|
> | DART-GUI-7B |21.67%|22.40%|8.48%|17.24%|
>
>
> Finally, regarding the scale of training tasks, we agree that 203 tasks are a starting point. While our results confirm that robust policies can be learned from this set, we recognize that broader task diversity is essential for mastering the vast space of user intents. We identify scaling up the diversity and quantity of training tasks as a critical direction for future work to investigate the scaling laws of GUI agents.
>
>
> [b] OpenCUA: Open Foundations for Computer-Use Agents.
>
> [c] UI-Vision: A Desktop-centric GUI Benchmark for Visual Perception and Interaction.

---

> ### Author Response · Authors · 2025-11-23
> **Response to Reviewer WyKM [3/4]**
>
> > **W3:** Inadequate failure mode analysis and limited mechanistic understanding: Section A.6 presents only two cherry-picked failure cases without systematic analysis. You may need to give some details, including:
>
> **A.** We appreciate the suggestion for a systematic failure analysis. To provide the requested mechanistic understanding, we conducted a comprehensive human study on 65 failed tasks (from DART-GUI-7B) to pinpoint limitations, and 55 corrected tasks (where DART succeeded but the baseline failed) to attribute specific improvements.
>
> We categorized the trajectories into the following failure modes:
>
> - **Reasoning Failures**: The agent correctly grounds UI elements but selects an incorrect action or logical sequence (e.g., clicking "Done" before changing a setting).
> - **Grounding Errors**: The agent fails to locate the target element or generates hallucinated coordinates.
> - **Looping/Stuck**: The agent enters a repetitive action sequence, or the environment remains trapped in a stagnant state (e.g., unclosable pop-ups or unresponsive pages) preventing progress.
> - **Knowledge Limitations**: The agent lacks the specific domain knowledge or application usage experience required to solve the task.
> - **Action Space Limitations**: The agent generates a valid intent, but the action fails due to API constraints or complex UI interactions (e.g., specific drag-and-drop mechanics).
>
>
> > **W3.1:** What percentage of errors are due to reasoning failures vs. grounding errors vs. action space limitations?
>
> **A.** We categorized the 65 failure cases of DART-GUI-7B into distinct error modes. The distribution is shown in **Table D**. We also analyzed the error distribution of the **55 tasks that were failed by the baseline but solved by DART** ("Corrected Tasks") to highlight where our method provides the most significant improvements.
>
> **Table D: Error Distribution Analysis**
>
> | Error Category | Baseline Failures (Corrected by DART) | DART-GUI-7B Failures (Remaining Errors) |
> | :--- | :---: | :---: |
> | **Reasoning Failures** | 61.82% | 40.00% |
> | **Grounding Errors** | 16.36% | 7.69% |
> | **Action Space Limitations** | 1.82% | 6.15% |
> | **Looping/Stuck** | 1.82% | 7.69% |
> | **Knowledge Limitations** | 18.18% | 38.46% |
>
>
> We also visualize the distribution shift in Figure A [https://dart-gui-iclr26.github.io/rebuttal/](https://dart-gui-iclr26.github.io/rebuttal/). The comparison reveals that DART significantly reduces **Reasoning Failures** (dropping from 61.82% in the corrected tasks to 40.00% in remaining errors) and **Grounding Errors** (16.36% to 7.69%). As the model becomes more capable of handling reasoning and grounding, **Knowledge Limitations** (38.46%) become the primary bottleneck for the remaining unsolved tasks.
>
> > **W3.2:** Which of the proposed methods (dynamic rollout, entropy filtering, etc.) are designed to address which specific failure modes?
>
> **A.** Dynamic Rollout reallocates sampling resources to tasks with lower success rates, which aids in mitigating Reasoning Failures by ensuring sufficient exploration for complex logical problems. Dynamic Trajectory Length encourages the agent to converge on more concise solutions; this tends to alleviate Grounding Errors, as shorter valid trajectories inherently reduce the cumulative probability of visual recognition mistakes. Entropy Filtering focuses updates on critical decision points with high uncertainty, stimulating the exploration of diverse reasoning pathways to further address Reasoning Failures.

---

> ### Author Response · Authors · 2025-11-23
> **Response to Reviewer WyKM [4/4]**
>
> > **W3.3:** How do these failure modes correlate with task complexity, trajectory length, or application type?
>
> **A.** We analyzed the failure modes against task metrics and application types (as shown in Figures A,B,C,D available at [https://dart-gui-iclr26.github.io/rebuttal/](https://dart-gui-iclr26.github.io/rebuttal/)[Anonymous URL]). Our statistical analysis reveals three key correlations:
>
> - *Correlation with Trajectory Length*: As shown in [Figure C](https://dart-gui-iclr26.github.io/rebuttal/), there is a positive correlation (0.29) between Looping/Stuck states and trajectory length, indicating that stuck agents tend to exhaust the maximum step limit. Notably, as shown in [Figure B](https://dart-gui-iclr26.github.io/rebuttal/), successful trajectories from DART-GUI-7B exhibit a remarkably even distribution across all length intervals (ranging consistently between 15% and 19%). In contrast, baseline failures are heavily skewed toward the maximum length (peaking at 31.2%), suggesting that DART maintains robust performance across both short and long-horizon tasks, whereas the baseline struggles to terminate effectively in complex scenarios.
> - *Correlation with Task Complexity*: As shown in [Figure C](https://dart-gui-iclr26.github.io/rebuttal/), Knowledge Limitations show a positive correlation (0.18) with task complexity. This suggests that as tasks become more intricate, the primary bottleneck shifts from basic operation to a lack of specific domain knowledge or functional understanding required to solve the problem.
> - *Correlation with Application Type*: [Figure D](https://dart-gui-iclr26.github.io/rebuttal/) shows that failure modes vary significantly by application interface . LibreOffice Calc, characterized by a dense grid of small cells, is dominated by Grounding Errors (100%), reflecting the difficulty in precise coordinate prediction. In contrast, applications like Chrome and LibreOffice Writer, which involve multi-step menu navigation and setting configurations, primarily exhibit Reasoning Failures (>75%), indicating that the challenge lies in logical planning rather than visual element identification.
>
>
>
> > **Q1:** Also, could you give out some trading off when scaling up the actors or interactive agents and going towards highly async environments,
>
> > **Q1.1:** what will happen,
>
> **A.** The primary trade-off lies in balancing GPU inference throughput (Actors) with CPU-intensive environment execution (Envs) to minimize resource idling. In our decoupled architecture (utilizing vLLM for high-throughput inference), scaling requires a proportional increase in resources to prevent system bottlenecks.
>
> Specifically, if the ratio is imbalanced, either the GPU idles (waiting for environment steps) or the environments stall (waiting for inference). Empirically, we found that 1 Nvidia H100 80G GPU (serving the Rollout Service) optimally supports 15–25 parallel OSWorld environments (each allocated 3 CPU cores). This ratio maximizes the utilization of both compute and simulation resources.
>
>
> > **Q1.2:** and how does your method prevent the risks of unstable training?
>
> **A.** In highly asynchronous settings, the primary risk is **policy staleness**, where the rollout policy lags behind the training policy, causing distribution shifts. DART mitigates this via two mechanisms:
>
> *   **Version-Aware Data Selection:** We strictly limit training data to trajectories generated by the current policy or its immediate predecessor. This bounds the policy lag, ensuring the behavior policy remains sufficiently close to the target policy without enforcing blocking synchronization.
>
> *   **Truncated Importance Sampling:** To mathematically correct for residual distributional mismatches (e.g., from quantization or slight policy lag), we apply a truncated importance weight $w$ to the gradient updates. Specifically, for a step with history $h$, state $s$, and action $a$:
>     $$ w = \min \left( \frac{\pi_{\theta_{\text{old}}}^{\text{Train}}(a|h,s)}{\pi_{\theta_{\text{old}}}^{\text{Rollout}}(a|h,s)}, C \right) $$
>     This weight rescales the policy gradient contribution, dampening high-variance updates caused by out-of-distribution samples and preventing model collapse.

---

### Author Response · Authors · 2025-12-03
**Quick Summary of Contributions and Rebuttal Updates**

Dear Area Chair,

We understand the significant workload resulting from the reassignment. To facilitate your review, we summarize our work (DART) and the major updates during the rebuttal:


**1. Reviewer Consensus**

We are grateful that the reviewers recognized the value of our work, particularly highlighting:

*   **Novel Architecture & Strong Engineering:** Our decoupled asynchronous architecture is considered a "strong engineering contribution" (`Reviewer WyKM`), "sophisticated" (`Reviewer NshA`), and a "novel design shift" (`Reviewer xFpq`) that addresses real bottlenecks in GUI agent training.
*   **Efficiency Gains:** The "transformative 5.5x improvement in environment utilization" (`Reviewer NshA`) and significant boosts in training throughput (`Reviewer xFpq`) were well received.
*   **Data Curation & Algorithm:** Our multi-level adaptive data curation is "comprehensive," "thoughtful" (`Reviewer WyKM`), and demonstrates a deep "understanding of learning dynamics" (`Reviewer NshA`).
*   **Performance & Impact:** The "thorough experimental validation" (`Reviewer WyKM`), "solid performance" on OSWorld (`Reviewer NshA`), and our commitment to open-sourcing (`Reviewer xFpq`) are recognized as significant contributions to the community.

**2. Key Rebuttal Updates**

Addressing reviewer comments, we have strengthened the paper with extensive new experiments, covering validation on unseen tasks, expanded baseline comparisons, and in-depth mechanism analysis, *etc*:

*   **Generalization:** (Response to Reviewer `WyKM` & `xFpq`) We clarified that our method has less risk of 'memorization', 'overfitting', or 'reward hacking', since the model is trained in a dynamic environment, and we use rule-based verification for reward. We then added three experiments to confirm this point. (1) We evaluated our model on newly constructed 171 human-annotated held-out tasks within the Ubuntu environment (distinct from the OSWorld tasks). (2) We compared our DART framework with the SFT scheme. (3) We evaluated our model on UI-Vision, a benchmark other than osworld. These experiments show that our method leads to good generalization capability across different tasks and domains.


*   **In-depth Comparisons:** (Response to Reviewer `WyKM` & `NshA`) We added conceptual comparisons with existing RL frameworks: DistRL and Sky-RL, and quantitative comparisons against GUI RL methods: ARPO and Zero-GUI. Our method is designed for long-horizon Desktop GUI tasks, effectively resolving unique VLM inference and training bottlenecks to support efficient and stable on-policy RL, achieving better performance on the OSWorld benchmark.


*   **More Experimental Analyses:** (Response to Reviewer `WyKM` & `xFpq`) We added robustness and ablation experiments to comprehensively analyse our method and each component. (1) We conducted additional experiments under a 30-step budget for fair and comprehensive benchmarking.  (2) We added full-benchmark ablations,  revealing that the decoupled framework alone contributes a +10.6% gain, while data curation adds the remaining +7.3%. (3) We added a detailed human study for failure mode analyses of the baseline and our model to show the reasoning and grounding improvements.

*   **More Implementation Details:** (Response to Reviewer `NshA` & `xFpq`) We added more details to the revised manuscript. (1) We show details on how the trajectory pool is constructed, how our method can lead to stable training, the resource consumption of our method, and our experimental setting, and (2) we include formal definitions of efficiency metrics and hyperparameter selection strategies to ensure reproducibility.

We believe these updates solidify the paper's contribution and address all concerns raised. We are committed to fully open-sourcing the framework and data.

Best regards,

The Authors

---

### Meta-Review · Area_Chair_WPwm · 2026-01-06

**Summary:**

This paper presents DART, a combination of systems and algorithm improvements aimed at efficiently training high performance GUI agents. The systems improvements center around removing synchronization bottlenecks and decomposing the various tasks in RL training into separate modules. Using DART, the authors training a state-of-the-art small open-source model for the OSWorld benchmark.

The primary concern remaining after the rebuttal is that there is a lack of benchmarking between the proposed system for multi-turn RL and existing systems. The author provided a high-level with Sky-RL, which is a good initial comparison, but a full benchmarking comparison is needed.

An empirical comparison is provided with ARPO and Zero-GUI, but these comparisons conflate the systems decisions with the algorithm and problem formulation decisions. A controlled comparison that isolates just the systems decisions is necessary to show how the systems decisions in DART improve upon those.

The authors also cite AREAL and ROLL. I encourage the authors to also compare with those directly in future versions of the paper. The claims about the efficiency of DART currently rely on an ablation of DART. This is useful to explain the performance of DART, but, it does not support the claims that DART enhances training efficiency for GUI agents in general.

**Reviewer Concerns:**

## Reviewer WyKM

- Limited algorithmic novelty, in particular compared to DistRL. I believe this concerned was partially addressed.
- Insufficient treatment of reward hacking. I believe this concern was addressed.
- Failure analysis and mechanistic understanding are lacking. I believe this concern was addressed.

## Reviewer NshA

- Lack of comparison with other high-performing models on OS-world. I believe this concern was addressed.
- Concerns about how the Experience Pool is defined. I believe this concern was addressed.
- Lack of benchmarking against other multiturn RL frameworks. I believe this concern is still outstanding.

## Reviewer xFqp

- The method may be highly tuned to the OS-world benchmark. I believe this concern was addressed.
- The results claiming improved sample efficiency are not from a clean comparison. I believe this concern was addressed.
- The ablations are not very well controlled. I believe this concern is still outstanding.
- Additional ablations are needed for experience pool and hyper-parameters. I believe this concerned was addressed.

**Reviewer Scores:**

I do not anticipate that reviewer scores would have meaningfully changed.

---

### Decision · Program_Chairs · 2026-01-26

Reject